# Functional and anatomical specificity in a higher olfactory centre

**Shahar Frechter[1]\*, Alexander Shakeel Bates[1], Sina Tootoonian[1,2,3], Michael-John Dolan[1,4], James Manton[1], Arian Rokkum Jamasb[5], Johannes Kohl[1], Davi Bock[4], Gregory Jefferis[1,5]\***

[1]Neurobiology Division, MRC Laboratory of Molecular Biology, Cambridge, United Kingdom; [2]Department of Neuroscience, Physiology and Pharmacology, University College London, London, United Kingdom; [3]Neurophysiology of Behaviour Laboratory, The Francis Crick Institute, London, United Kingdom; [4]Janelia Research Campus, Howard Hughes Medical Institute, Chevy Chase, United States; [5]Department of Zoology, University of Cambridge, Cambridge, United Kingdom

**Abstract** Most sensory systems are organized into parallel neuronal pathways that process distinct aspects of incoming stimuli. In the insect olfactory system, second order projection neurons target both the mushroom body, required for learning, and the lateral horn (LH), proposed to mediate innate olfactory behavior. Mushroom body neurons form a sparse olfactory population code, which is not stereotyped across animals. In contrast, odor coding in the LH remains poorly understood. We combine genetic driver lines, anatomical and functional criteria to show that the *Drosophila* LH has ~1400 neurons and >165 cell types. Genetically labeled LHNs have stereotyped odor responses across animals and on average respond to three times more odors than single projection neurons. LHNs are better odor categorizers than projection neurons, likely due to stereotyped pooling of related inputs. Our results reveal some of the principles by which a higher processing area can extract innate behavioral significance from sensory stimuli.
DOI: https://doi.org/10.7554/eLife.44590.001

**\*For correspondence:**
frechter@mrc-lmb.cam.ac.uk (SF);
jefferis@mrc-lmb.cam.ac.uk (GJ)

**Competing interests:** The authors declare that no competing interests exist.

## Introduction

In thinking about the transition from stimulus through perception to behavior, chemosensory systems have become increasingly studied due to their relatively shallow architecture: just two synapses separate the sensory periphery from neurons that are believed to form memories or instruct behavior (*Wilson and Mainen, 2006*; *Masse et al., 2009*). However, there are many shared organizational features with other sensory systems : for example, integration of information originating from distinct sensory receptors, and multiple levels of parallel and hierarchical processing. It is therefore possible that detailed studies of chemosensory systems may reveal general principles relevant to neurons that are considerably deeper in other sensory modalities. This study uses single cell neuroanatomy and genetically targeted *in vivo* electrophysiology to addresses two principal questions in the context of the *Drosophila melanogaster* olfactory system. First, can a higher olfactory center encode odors in a stereotyped way? More specifically, are there distinct cell types with reproducible odor responses across animals? Second, what is the nature of the population code in a higher olfactory area linked to innate behavior? How might the code relate to the behavioral requirements of the animal and how does it differ from neurons underlying learned responses?

The olfactory systems of mammals and insects have many similarities, including the presence of glomerular units in the first olfactory processing center. Second order neurons then make divergent projections onto multiple higher olfactory centers. For example in both flies and mice there are separate projections to areas proposed to be specialized for memory formation (mushroom body and

piriform cortex, respectively) and unlearned olfactory behaviors (lateral horn and e.g. cortical amygdala) (*Heimbeck et al., 2001*; *Jefferis et al., 2007*; *Sosulski et al., 2011*; *Root et al., 2014*). In insects in general and *Drosophila melanogaster* in particular, the anatomical and functional logic of odor coding in the mushroom body (MB) and its relationship to olfactory learning has been intensively studied. About 150 projection neurons (PNs) relay information to the input zone of the MB, the calyx, where they form synapses with the dendrites of ~2000 Kenyon cells, the intrinsic neurons of the MB (reviewed in *Masse et al., 2009*). There is limited spatial stereotypy in these projections (*Jefferis et al., 2002*; *Wong et al., 2002*; *Tanaka et al., 2004*; *Jefferis et al., 2007*; *Lin et al., 2007*) and each KC receives input from an apparently random sample of ~6 PNs (*Caron et al., 2013*). KC axons form a parallel fibre system intersected by the dendrites of 35 MB output neurons (*Aso et al., 2014a*); it is proposed that memories are stored by synaptic depression at these synapses. KC odor responses are very sparse (*Perez-Orive et al., 2002*; *Turner et al., 2008*), which is proposed to minimize interference between different memories, but do not appear stereotyped across animals (*Murthy et al., 2008*, but see *Wang et al., 2004*). Random PN-KC connectivity is consistent with the idea that KC responses acquire meaning through associative learning rather than having any intrinsic valence. In mice, pyramidal cells of the piriform cortex also integrate coincident inputs from different glomeruli (*Miyamichi et al., 2011*; *Davison and Ehlers, 2011*), but as in the MB the available evidence suggests that this integration is not stereotyped from animal to animal (*Stettler and Axel, 2009*; *Choi et al., 2011*).

In contrast to extensive studies of the MB, there is much more limited information concerning anatomy and function of the lateral horn (LH); this is also true for higher olfactory centers of the mammalian brain that have been hypothesized to serve a similar functional role such as the cortical amygdala. Studies examining the axonal arbors of PNs showed that these have more stereotyped projections in the LH than in the MB (*Marin et al., 2002*; *Wong et al., 2002*; *Tanaka et al., 2004*; *Jefferis et al., 2007*). Since at least three classes of LH neurons (LHNs) appeared to have dendrites in stereotyped locations (*Tanaka et al., 2004*; *Jefferis et al., 2007*), it was hypothesized that these neurons have stereotyped odor responses that are conserved from animal to animal. The first studies of odor responses of LHNs focussed on pheromone responses of neurons suspected to be sexually dimorphic in number or anatomy owing to their expression of the *fruitless* gene (*Ruta et al., 2010*; *Kohl et al., 2013*). *Kohl et al. (2013)* characterized three neuronal clusters showing that they responded in a sex- and class-specific manner and ranged from narrowly tuned pheromone-specialists to more broadly responsive neurons.

Pheromone responsive second order neurons project axons to a specialized subregion of the LH (*Seki et al., 2005*; *Jefferis et al., 2007*) and the extent to which pheromone responsive LHNs are typical of the whole LH has been questioned by other studies. In particular (*Gupta and Stopfer, 2012*) recorded from a random sample of neurons in the locust LH, reporting that all LHNs were extremely broadly tuned and without finding evidence of neurons with repeated odor profiles; they eventually concluded that generalist LHNs are unlikely to be stereotyped encoders of innate behavior. *Fişek and Wilson (2014)* carried out the first electrophysiological recordings of non-pheromone LH neurons in *Drosophila*. They recorded from two genetically identified cell types with reproducible response patterns: one LHN class responded to 1 out of 8 tested odorants, the other responded to all odorants. Although these results suggested that generalist LHNs can also have stereotyped odor responses, the limited number of neurons investigated precluded general conclusions about LH odor coding. Studies of analogous regions in the mammalian brain are even more challenging, but recent recordings by *Lurilli and Datta (2017)* from the cortical amygdala found no evidence for response stereotypy or encoding of the behavioral valence or chemical category of odors.

We have taken a stepwise approach to understanding the organizational and functional logic of the LH. We first reasoned that it was essential to characterize its cellular composition and to develop approaches for reproducible access to different cell populations. We achieved this by screening and annotating genetic driver lines, hierarchically classifying single cell morphologies, and using whole brain electron microscopy (EM) data to place rigorous bounds on total cell numbers, which turn out to be much greater than anticipated.

We then used genetically targeted single cell electrophysiology and anatomy to reveal principles of odor coding in genetically defined cell populations, finding that LHNs typically respond to more odors but with fewer spikes than their PN input. Since we found that LHNs show stereotyped odor responses across animals, we carried out a detailed analysis of neuronal cell types. We show that

functional and morphological criteria can both be used to define cell type and that they are highly consistent. We then show that LHNs are better odor categorizers than their PN inputs, providing one justification for their distinct coding properties. Finally we use EM data to measure direct convergence of different olfactory channels onto both local and output neurons of the LH, providing an anatomical substrate for the response broadening. Taken together these results reveal some of the logic by which the nervous system can map sensory responses onto behaviorally relevant categories encoded in a population of stereotyped cell types in a higher brain area.

## Results

### Key anatomical features of the lateral horn

Our key goal in this study was to understand the coding principles of third order neurons underlying innate olfactory behaviors. Nevertheless it is hard to understand the functional properties of a brain area without a basic understanding of the number and variety of constituents neurons. We used a wide variety of experimental/analytic approaches to obtain a comprehensive overview of the functional anatomy of the LH. We now present the observations most relevant to odor coding, organized hierarchically. We present further details in Materials and methods and online supplements including accompanying 3D data (jefferislab.org/si/lhlibrary).

Neuropil volume is indicative of the energetic investment in particular sensory information (*Sterling and Laughlin, 2015*), and strongly correlated with length of neuronal cable and synapse numbers (e.g. *Schlegel et al., 2017*). We find that the volume of the LH is 70% of the volume of the whole MB. However many LHNs extend axons beyond the LH; we estimate that the total volume of LHN arbors is therefore actually (40%) greater than the MB. This large investment in neuropil volume argues for the significance of the LH in sensory processing, whereas the literature currently contains 13 fold more studies of the *Drosophila* MB than LH (Materials and methods section Neuropil Volumes).

The number of neurons in a brain area is a key determinant of neuronal coding. A classic EM study cutting the MB's parallel axon tract counted 2200 Kenyon cells (*Technau and Heisenberg, 1982*), while comprehensive genetic driver lines contain up to 2000 KCs (*Aso et al., 2009*). However the number of LHNs has remained undefined, since there is no single tract to cross-section, nor any driver line labelling all LHNs. In the locust, (*Gupta and Stopfer, 2012*) estimated that there are fewer LHNs than PNs. We combined light level image data with whole brain electron microscopy (EM) (*Zheng et al., 2018*) to address this question. Our anatomical screen (see below) identified 31 primary neurite tracts entering the LH (*Figure 1B* and *Table 1*); a random EM tracing procedure targeting 17 of the largest tracts yielded an estimate of 1410 LHNs (90% CI 1368–1454, see Experimental Procedures). Each tract consists predominantly of either output or local neurons giving an estimate of about 580 LH local neurons (LHLNs, 40%) and 830 LH output neurons (LHONs, 60%). These results show that LHONs are much more numerous than second order input PNs and within a factor of 2 of the number of third order MB Kenyon cells. The large number of KCs enables sparse odor coding, which is proposed to avoid synaptic interference during memory formation (reviewed by *Masse et al., 2009*). Why should the LH also contain so many neurons?

### Driver lines and hierarchical naming system for LHNs

Transgenic driver lines are the standard approach to label and manipulate neurons in *Drosophila* (*Venken et al., 2011*). Given the large number of LHNs, it seemed essential to identify lines targeting subpopulations to test our hypothesis that LHNs are stereotyped odor encoders. When we began our studies, the only relevant lines came from *Tanaka et al. (2004)*, who identified four drivers labeling distinct populations of LHNs from a screen of ~4,000 GAL4 lines. We carried out an enhancer trap Split-GAL4 screen (*Luan et al., 2006*), eventually selecting 234 lines containing LHNs from 2769 screened (see Experimental Procedures for details). These lines gave us access to the majority of known classes of LHNs and were used for most functional studies in this paper. However they were rarely cell type-specific, making them less suitable for behavioral experiments.

In addition to the previous absence of genetic reagents, LHNs are morphologically extremely diverse and do not innervate discrete glomeruli or compartments (*Laissue et al., 1999*; *Tanaka et al., 2008*; *Aso et al., 2014a*). For all these reasons, there was no prior system to name

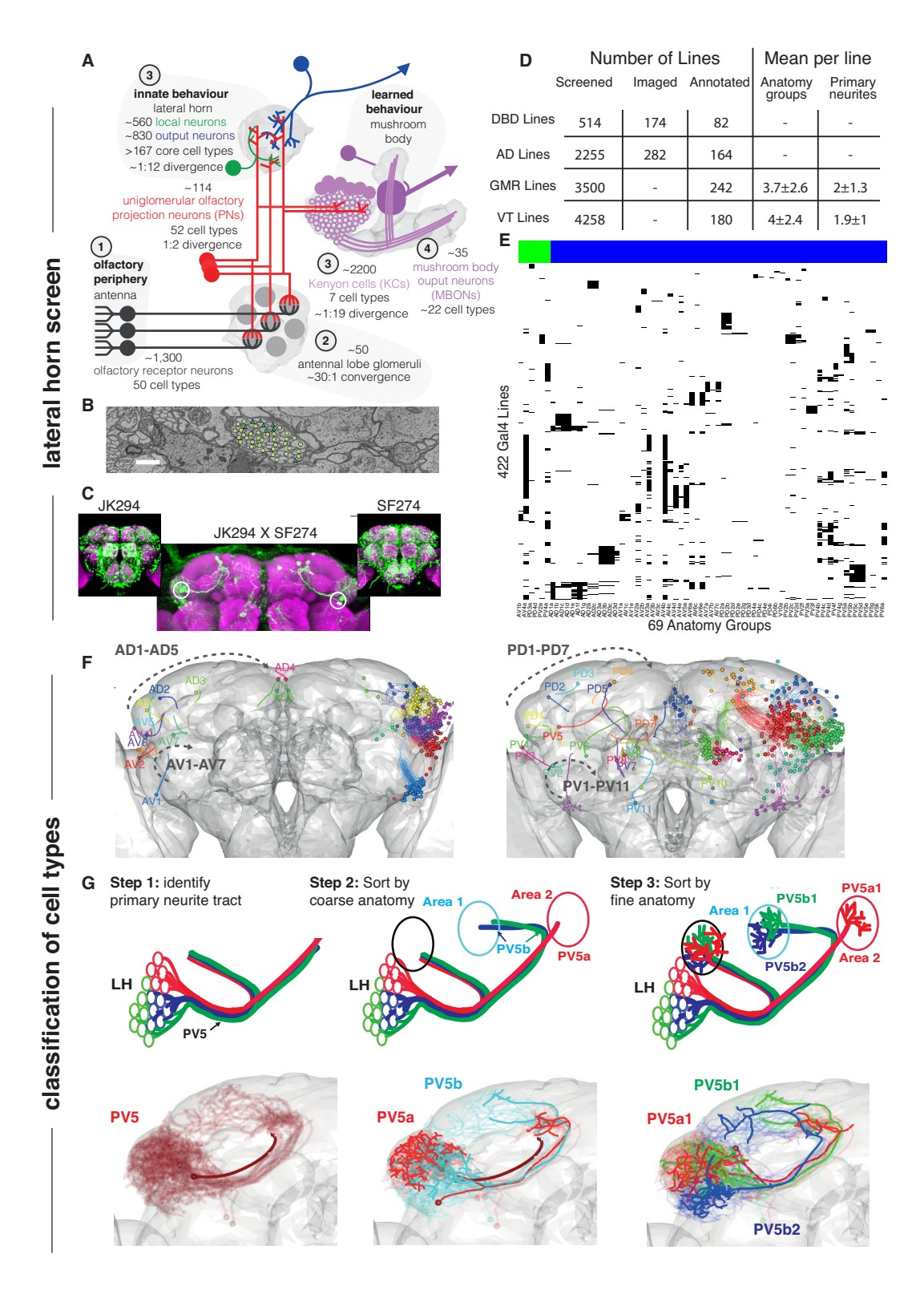

**Figure 1.** Screen for genetic driver lines labeling Lateral Horn Neurons. (A) Flow diagram following olfactory information to third order neurons of the LH and MB calyx. (B) Section through the PV5 primary neurite tract within the EM data; each profile was identified as an LHN (circles) or non LHN (triangles) by tracing to the first branch point. Scale bar 1 μm. (C) Sample Split-GAL4 intersection with parental lines inset. Cell body locations marked with white circle. (D) Summary table for the genetic screen. (E) Matrix summarizing which driver lines contained which anatomy groups (colored

*Figure 1 continued on next page*

*Figure 1 continued*

LHON = blue, LHLN = green). The vast majority of genetic driver lines labeled only a few LH anatomy groups. NB these data are also available as a supplementary spreadsheet. (**F**) Anterior (left) and posterior (right) views of the different LHN primary neurite tracts demonstrating the broad origin of LHNs. Grey dashed arrow indicates the order of increasing tract numbers for ventral PNTs (Lower arrow) and dorsal PNTs (above). The entry point into the brain rather than soma location was the point of reference for naming tracts. (**G**) Upper panels, cartoons summarizing the logic of the LHN naming system, lower panels, the PV5 primary neurite as an example. Note this includes only 3 out of 24 cell types in PV5. For each cell type one cell is highlighted (thick lines).

DOI: https://doi.org/10.7554/eLife.44590.002

The following figure supplement is available for figure 1:

**Figure supplement 1.** Summary of anatomical and functional screen.

DOI: https://doi.org/10.7554/eLife.44590.003

and classify different LHNs. We devised a hierarchical naming system with three levels of increasing anatomical detail to disambiguate neurons (*Figure 1F and G*): (1) *primary neurite tract*: the tract containing fibers connecting the soma to the rest of the neuron, (2) *anatomy group*: neurons sharing a common axon tract and broadly similar arborizations in the LH and target areas, (3) *cell type*: the finest level revealed by reproducible differences in precise axonal or dendritic arborization patterns. We use single cell data to illustrate these three levels for three closely related cell types (PV5a1, PV5b1, PV5b2) (*Figure 1G*, see Materials and methods for details). This system was key to successful genetic screening as well as planning and reporting functional experiments.

**Table 1.** LHN tracts characterized in electron microscopy data.

Tracts match the Primary Neurite Tract nomenclature defined in *Figure 1*. Type indicates whether the tract contains output or local neurons or a mix of both. Profiles indicates the total number of profiles within the tract. Est.LHNs indicates the sampling based estimate for the number of LHNs in the tract. Range gives a 90% confidence interval.

| Tract | Type | Profiles | Est. LHNs | Range | Recorded |
|---|---|---|---|---|---|
| AV4 | LHLN>>LHON | 324 | 252 | 244–259 | Yes |
| PV4 | LHLN>LHON | 158 | 155 | 152–158 | Yes |
| PV2 | LHLN>>LHON | 193 | 92 | 81–102 | Yes |
| PD3 | LHLN | 75 | 59 | 43–75 | |
| PD4 | LHLN | 88 | 22 | 10–33 | |
| | –LHLNs– | 838 | 578 | 555–602 | |
| AV3 | LHON>LHLN | 144 | 140 | 140 | |
| PD2 | LHON | 193 | 128 | 128 | Yes |
| PV5 | LHON | 127 | 119 | 119 | Yes |
| AD1 | LHON | 286 | 116 | 102–130 | Yes |
| AV6 | LHON | 323 | 106 | 96–115 | Yes |
| AV2 | LHON>>LHLN | 98 | 63 | 49–77 | Yes |
| AD3 | LHON | 59 | 59 | 59 | |
| AV7 | LHON | 141 | 48 | 25–70 | |
| AV1 | LHON | 33 | 25 | 25 | |
| AV5 | LHON | 108 | 17 | 7–27 | |
| PV3 | LHON | 52 | 12 | 0–25 | |
| AD2 | LHON | 52 | 0 | 0 | |
| | –LHONs– | 1616 | 832 | 797–868 | |
| | –Total– | 2454 | 1411 | 1368–1454 | |

DOI: https://doi.org/10.7554/eLife.44590.004

Building on our initial screen, we also annotated (*Figure 1D*) the widely used FlyLight (often referred to as GMR lines, *Jenett et al., 2012*) and Vienna Tiles libraries (VT lines, *Tirian and Dickson, 2017*). These lines are now very widely used in *Drosophila* neurobiology, in part because co-registered 3D image data are publicly available (e.g. through virtualflybrain.org *Milyaev et al., 2012*; *Manton et al., 2014*). The vast majority of genetic driver lines labeled only a few LH anatomy groups (mean of 3.8) while just 21/422 lines contained more than 8; we did not find any lines specific to multiple LHN anatomy groups without labelling other neurons in the central brain (*Figure 1D and E*). Similarly, none of the primary neurite tracts proved LH specific, although some were highly LH enriched (*Figure 1B*). This demonstrates how hard it is to obtain LH selective lines that label most or even a large portion of the LHNs. At the conclusion of our screen we had identified 69 distinct LHN anatomy groups – that is, neurons with substantially different axonal tracts/arborisation patterns – each of which was consistently labeled by a subset of driver lines. This cellular and genetic diversity significantly exceeded our initial expectations and represented an almost order of magnitude increase over prior studies. It also contrasts very strongly with the seven genetically defined Kenyon cell types comprising the third order neurons of the mushroom body (*Aso et al., 2014a*).

This second screen of GMR/VT lines provides a link between our LHN classification and experimentally valuable resources including further driver lines and co-registered 3D image data (see Materials and methods). Indeed, building on these annotations we went on to prepare a large collection of highly specific intersectional Split-GAL4 lines selectively targeting specific LH cell types; this facilitates many experiments including behavioral analysis for which our first generation split-GAL4 reagents were less suitable (see *Dolan et al., 2019*, sister manuscript).

## Single cell anatomy of the lateral horn

The results presented so far provide principled estimates of the number of LHNs, identify genetic reagents for their study and develop a hierarchical nomenclature classification system. The final part of our neuroanatomical groundwork was to carry out a large scale single cell analysis of the LH in order to gain an initial understanding of the variety of cell types that it contains.

Given our new estimate that there are ~1400 neurons LHNs, what is the anatomical and functional diversity amongst this large number of neurons? To address this, we co-registered FlyCircuit neurons (*Chiang et al., 2011*) and neurons recorded during this study, segmenting each neuron into predicted axonal and dendritic domains (*Figure 2*, see Materials and methods). We created an online 3D atlas of 1619 LHNs as well as 1258 LH input neurons. We first reviewed LH inputs (*Figure 2A*). The principal uniglomerular olfactory inputs to the LH have been well-studied but we found 26 new classes including many non-olfactory inputs (see *Figure 2—figure supplement 1A* and Materials and methods for details). Multiglomerular olfactory neurons, and presumptive thermosensory, hygrosensory, and mechanosensory neurons were all concentrated in a ventromedial domain of the LH (*Figure 2B*), emphasizing that the LH is a multimodal structure.

To classify LHNs, i.e. neurons with presumptive dendrites in the LH, we assigned neurons to anatomy groups and cell types (*Figure 1G*) using NBLAST clustering (*Costa et al., 2016*), followed by a close manual review of within and across cell type stereotypy in fine branching patterns (*Figure 2—figure supplement 1B*). We found that there is no unique statistical definition (i.e. single cut height for NBLAST clustering) that is appropriate for all LH cell types (*Figure 2—figure supplement 1B′*), even when these anatomical cell type can be validated by other cellular properties (see below). Nevertheless for those cell types with more than one neuron, NBLAST identified the correct, manually ascribed cell type ~80% of the time (*Figure 2—figure supplement 1E*).

We identified a total of 261 LHN cell types divided into 34 local cell types (LHLNs) with arbors restricted to the LH and 227 LHON cell types with axons beyond the LH. Most cell types originate from the tracts identified by EM as containing the largest number of LHNs (*Table 1*). LHLNs are associated with about half the number of tracts as LHONs (i.e. 5 vs 12 major EM tracts, *Table 1*). Similarly the AV4 and PV4 tracts account for 73% of the estimated 580 local neurons, whereas it takes 5 LHON tracts to reach this proportion.

LHONs are more anatomically diverse, originating from 29/31 tracts as well as having a wide range of axonal projections. Some LHNs clearly had dendrites in multiple neuropils. We therefore focussed on a set of 134 *core* LHON cell types with >50% their dendrites within the LH (*Figure 2—figure supplement 1F*; see Materials and methods for details). These LHONs project to a wide array of target areas (*Figure 2E*). The superior protocerebral neuropils (SLP, SMP and SIP) are the most

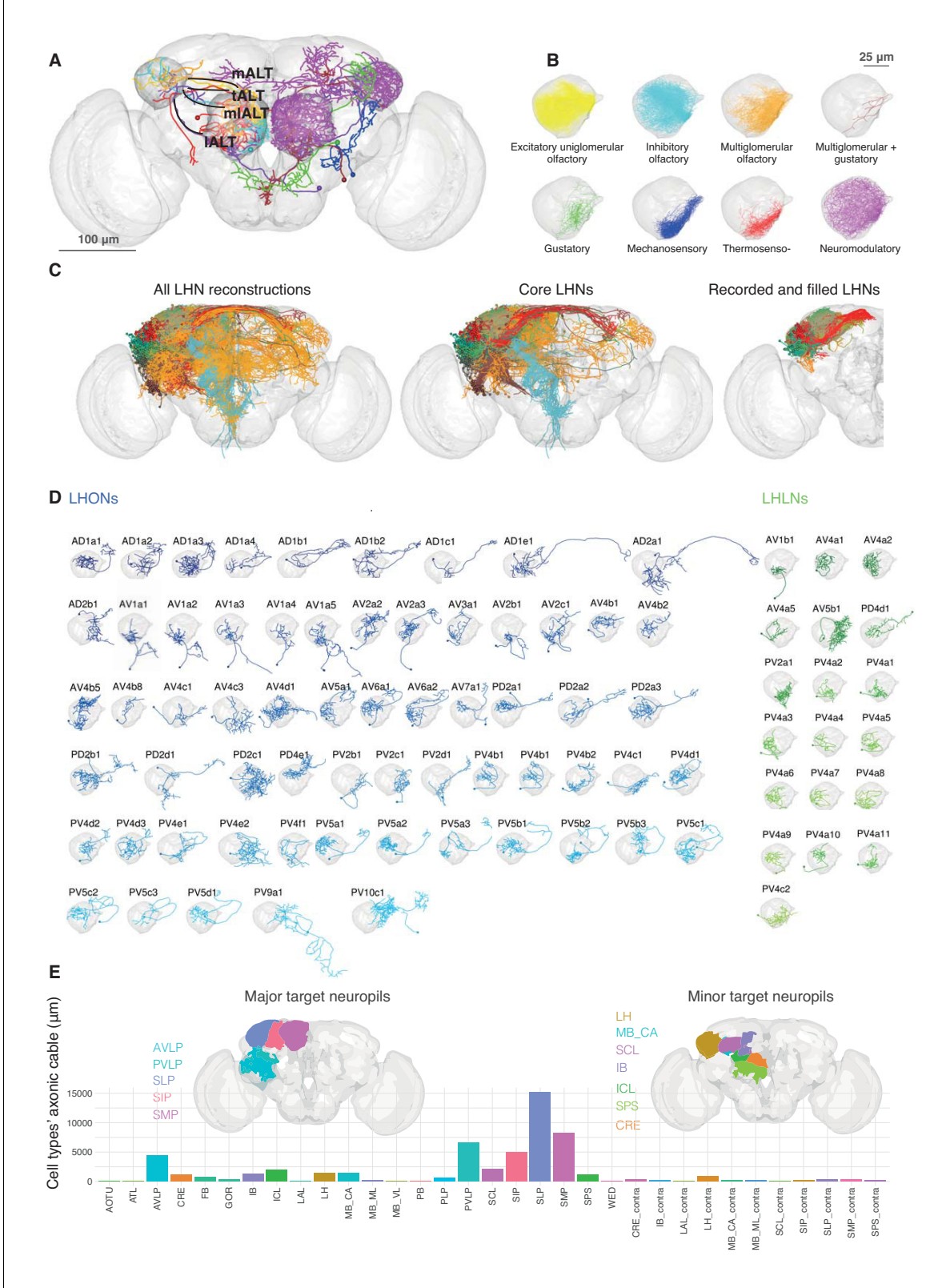

**Figure 2.** Single cell anatomy of the Lateral Horn. (**A**) Sample single projection neurons with axonal projections in the LH, showing all major axon tracts and sensory modalities that provide input. (**B**) Close up of the LH with axonal arbors for all FlyCircuit neurons of each presumptive sensory modality. (**C**) Overview of our annotated LHN skeleton library showing all skeletons with LH arbors, core LHN cell types (see *Figure 2—figure supplement 1*) and those neurons reconstructed after electrophysiological recording in the present study. Neurons colored by anatomy group. (**D**) Visualization of single

*Figure 2 continued on next page*

*Figure 2 continued*

exemplars for all cell types for which we have >=3 skeletons in the library, or from which we made electrophysiological recordings in this study. Output neurons in blue, local neurons in green. (E) Bar chart showing, for each target neuropil, the total axonal cable length contributed by all core LHONs (calculated as sum of mean for each identified cell type). Brains plots show in major (> 3 mm axonal cable) and minor (1–3 mm) targets of LHONs. Brain neuropil according to *Ito et al. (2014)*; mALT, medial antennal lobe tract, tALT, transverse antennal lobe tract, mlALT, medio-lateral antennal lobe tract, lALT, lateral antennal lobe tract.

DOI: https://doi.org/10.7554/eLife.44590.005

The following figure supplements are available for figure 2:

**Figure supplement 1.** Summary of neuron skeleton data for LHNs and PNs.

DOI: https://doi.org/10.7554/eLife.44590.006

**Figure supplement 2.** Local vs. Output AV4 and PV4 clusters.

DOI: https://doi.org/10.7554/eLife.44590.007

extensively innervated; they are the location of 'convergence zones' where direct olfactory output from the LH may be integrated with learned olfactory information from the mushroom body (*Aso et al., 2014a*). The ventrolateral protocerebrum (AVLP, PVLP) is the next major target; this area also receives extensive input from visual projection neurons originating in the optic lobes (e.g. *Panser et al., 2016*) and is innervated by dendrites of descending neurons, including at least two now known to be downstream of LHONs (*Huoviala et al., 2018*). Some LHONs also have both axonal and dendritic domains in the LH. We noticed that few cell types project to the contralateral hemisphere, perhaps because most olfactory projection neurons in the adult fly already receive information from both antennae.

The most studied projections to the LH are GH146-GAL4 positive, excitatory uniglomerular projection neurons from the antennal lobe (AL) that run through the medial antennal lobe tract (mALT). 3D atlases of these uniglomerular PNs have been constructed previously based on co-registration and annotation of single cell data (*Jefferis et al., 2007*; *Costa et al., 2016*). Nevertheless, there are numerous additional inputs to the LH. We annotated LH input neurons from FlyCircuit and divided them into 41 different groups based on the axon tract they use to reach the LH and their pattern of dendritic arborization (*Figure 2—figure supplement 1A-B*), which we used as a proxy for the modality of the sensory information they encode. We extended the naming system of *Tanaka et al. (2012a)* to include 26 types not previously identified (see Materials and methods). *Tanaka et al. (2012a)* have described five mALT types, three mediolateral antennal lobe tract (mlALT) types, three lateral antennal lobe tract (lALT) types, and three transverse antennal lobe tract (tALT) PN types that project to the LH from the AL. PNs taking any tract can have uniglomerular, multiglomerular or non-glomerular dendritic arborisation in the AL, sampling broadly or sparsely from the available odor channels. Unlike *Tanaka et al. (2012a)*, but as has been observed in the larva (*Berck et al., 2016*), we find that some of these mALT olfactory projections do not arborize in the MB calyx (data not shown). GABAergic olfactory input is known to be supplied via PNs traversing the mlALT (*Wilson and Laurent, 2005*; *Okada et al., 2009*), whereas the majority of projections through the mALT and lALT are thought to be cholinergic (*Tanaka et al., 2012a*). We were not able to find a few PN types that had been described in the literature to project to the LH, including AL-MBDL (*Tanaka et al., 2012a*).

Input distribution is not uniform within the LH. Excitatory uniglomerular and inhibitory GABAergic PNs project widely but spare a ventromedial stripe of the LH, which is the focus of multiglomerular olfactory neurons and other sensory inputs; this same arborization pattern is shared by two neuromodulatory neurons releasing octopamine (*Busch et al., 2009*) and serotonin (*Roy et al., 2007*). In contrast, excitatory multiglomerular projection neurons are heavily concentrated in the ventromedial LH, where their arbors intermingle with PNs from thermosensory and hygrosensory glomeruli (*Frank et al., 2015*; *Frank et al., 2017*); in addition undescribed projection neurons from the Wedge neuropil may carry mechanosensory wind input (*Yorozu et al., 2009*; *Patella and Wilson, 2018*). Gustatory projection neurons also innervate this domain, although these are concentrated in an anterior-medial domain adjacent to the LH (see also *Kim et al., 2017*). In conclusion, the ventral LH receives multimodal input and likely to be involved in multimodal integration while the remainder is predominantly olfactory. Due to sampling biases in the FlyCircuit dataset, it is very likely that some cell types are over-represented, while others may be missing altogether. It has been reported from

electron microscopy that the mALT contains ~288, the mlALT 88–100 and the tALT ~60 fibers from the vicinity of the AL (*Tanaka et al., 2012b*).

In order to enable more effective exploration of these data, we have prepared a number of downloadable data and source code resources (see Online resources). These include a 3D enabled web application at jefferislab.org/si/lhlibrary which also links to the highly selective split-GAL4 reagents described in our (*Dolan et al., 2019*) sister manuscript as well as cross-referencing 26 LHN cell types recently characterized by *Jeanne et al. (2018)*.

## Odor responses of lateral horn neurons

The neuroanatomical groundwork that we have just summarized includes a huge amount of detail that will be relevant for many circuit studies. However as we turn our attention to olfactory coding one question, one major question stands out. Why are there so many LHN cell types? To answer this question we began by defining the odor response properties of LHNs and comparing them with their presynaptic partners, the PNs. We also hoped to contrast LHN responses with those of MB Kenyon cells, the other main class of third order olfactory neuron.

With genetic driver lines in hand we were able to carry out targeted recordings from LHNs (*Figure 3*). Given that these cells had unknown response properties and our previous experience was that calcium signals in LHN somata are not a sensitive measure of LHN firing, we carried out in vivo whole cell patch clamp recordings as we have previously described (*Kohl et al., 2013*). We recorded 587 cells of which 410 (242 LHONs, 84 LHLNs, and 84 identified PNs) reached the criteria for inclusion in our population analysis (see Experimental Procedures). Comparing basic electrophysiological parameters across different groups, both LHONs and LHLNs generally have a much higher input resistance and lower cell capacitance than PNs (*Figure 4D* and *Figure 3—figure supplement 1D*); this suggests that the energetic costs of individual spikes will be lower in LHONs than PNs.

We selected an odor set designed to excite many different olfactory channels (*Hallem and Carlson, 2006*; *Münch and Galizia, 2016*) that included diverse chemical groups including acetates, alcohols, organic acids, aldehydes, ketones, amines and phenyls (*Figure 4B*). Our core odor set consisted of 36 odors although up to 53 odors were used for some cells in the study. LHONs generally showed little spiking in the absence of odor with a mean firing rate of 0.1 Hz. In contrast PNs showed a higher mean baseline firing rate of 1.4 Hz (*Figure 3* and *Figure 4C*) consistent with previous reports (e.g. *Wilson et al., 2004*; *Jeanne and Wilson, 2015*). The baseline firing rate of LHLNs was intermediate with a firing rate of 1 Hz. Odor responses were reliable for all three groups and it was rare for a cell to respond to one odor presentation without responding to the other presentations of the same odor (*Figure 4H*). Cell averaged single trial response reliability was slightly higher for LHONs. This probably reflects the higher baseline firing rate of PNs since when we considered only stronger responses, reliability for all groups approached 100% (*Figure 3—figure supplement 1E*).

If we consider every single odor presentation, the mean firing rate was similar across PNs, LHLNs, and LHONs (4.4–5.2 Hz). However PNs (and LHLNs) responded to fewer odors then LHONs: 12% of odors elicited a significant PN response compared to 35% for LHONs (*Figure 4B,E*; see Materials and methods for definition of a significant response). Consistent with this, PN responses were sparser than LHONs (*Figure 4F,G*). If we consider significant excitatory odor responses only, we see that when an LHON responds to an odor, it does so with a lower firing rate: 21 Hz for PNs and 14 Hz for LHONs (*Figure 4C*).

In conclusion, LHONs are on average 10x quieter than PNs at baseline, show significant responses to 3x more odors, but have lower evoked firing rates, consequently firing a similar total number of spikes.

## Defining functional cell types

Our recordings indicated that cells fall into distinct groups based on their odor tuning profile (*Figure 5*). For example, morphologically similar neurons belonging to the same anatomy group could be subdivided by their odor-evoked responses. Close inspection revealed subtle morphological differences between these subgroups (*Figure 5B,E*). However when looking over the entire cell repertoire it was evident that although we used a large odor set, many cells that were anatomically

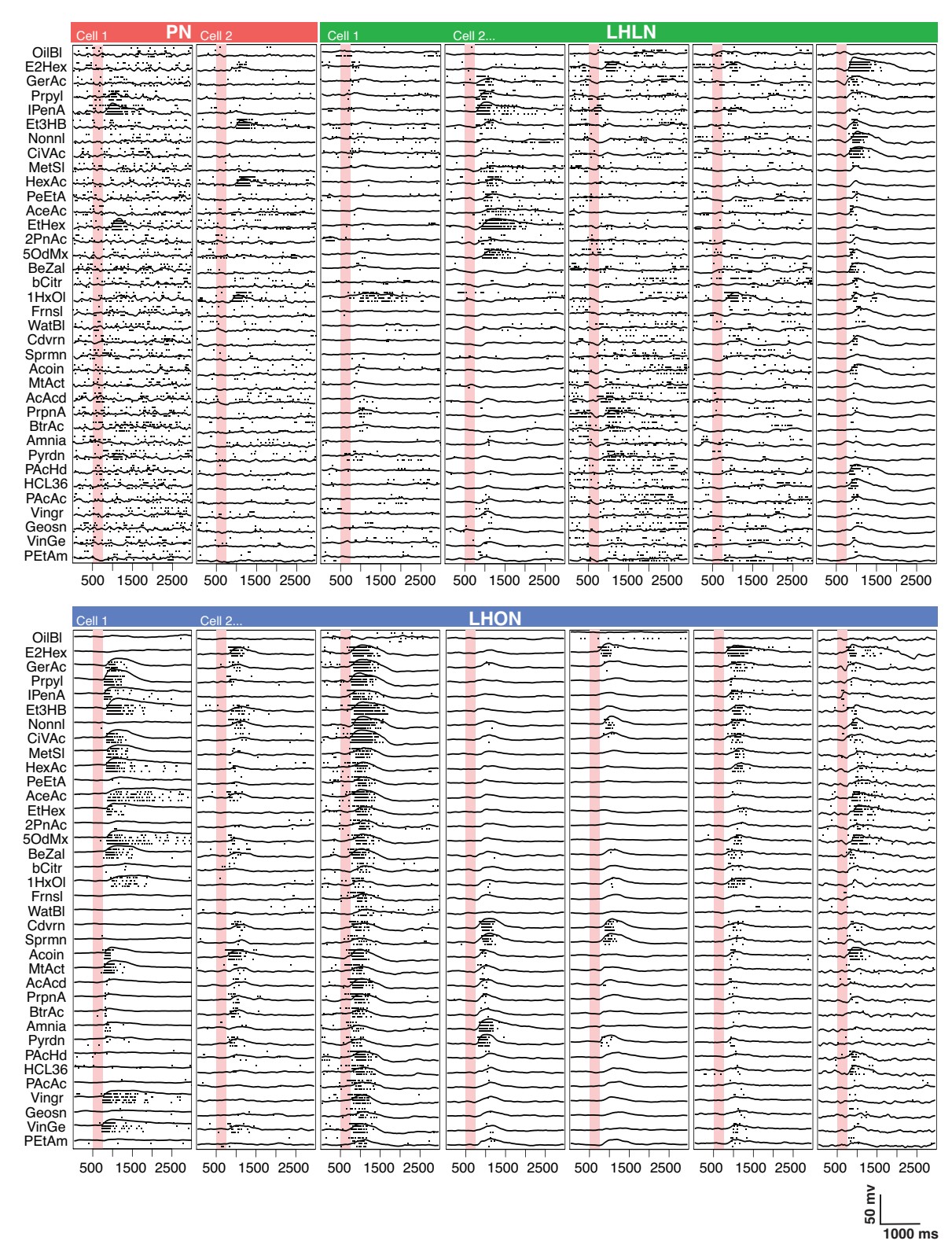

**Figure 3.** Comparing odor responses of second and third order olfactory neurons. Raster plots for two PNs (red), five LHLNs (green), and seven LHONs (blue). Each odor was presented 4 times to each cell with a 250 ms valve opening starting 500 ms after recording (red bar). For each odor the voltage response of the four trials was averaged (continuous line) while rasters show the spiking response for each presentation. Note the progressive reduction

*Figure 3 continued on next page*

*Figure 3 continued*

in baseline firing rate and sparseness between PNs and LHLNs, and LHONs. Odors abbreviated on the left can be identified from a supplementary spreadsheet.

DOI: https://doi.org/10.7554/eLife.44590.008

The following figure supplement is available for figure 3:

**Figure supplement 1.** Cell Physiolgical Parameters.

DOI: https://doi.org/10.7554/eLife.44590.009

completely distinct, had rather similar odor responses that we could not reliably separate by automated analysis (*Figure 5C,F*).

Although it would have been desirable to assign a *functional cell type* based solely on odor response data, the factors that we have just outlined made this impossible in practice. We therefore used a two stage process. The first stage was to use coarse anatomical features (primary neurite and axon tract) to separate the cells into anatomy groups. The second stage then divided cells within an anatomy group based on their odor response properties (*Figure 5A*). This finest level of classification into distinct cell types therefore depended solely on their functional properties. This classification was initially carried out manually and identified 64 functional cell types, of which 59 contain two or more exemplars and 42 contain 3 or more.

While some functional cell types had low variability in odor responses and were clearly segregated, others were less easily classified. In particular, we regularly observed that the common odor response profile of cells that we eventually assigned to the same functional cell type, was masked by differences in response magnitude or threshold (*Figure 5D*). This variability may originate from differences in the number or strength of inputs to that cell type within or across animals. Given that we recorded from one cell per animal, we cannot exclude variation due to experimental factors (e.g. small differences in fly position or orientation, quality of recording and general state of the fly) that may affect the response strength. However we did not find any consistent relationship between cell-recording parameters (cell capacitance, membrane resistance and pipette resistance) and the strength of the response, suggesting that this is not an artifact of recording conditions (*Figure 3—figure supplement 1*). Furthermore recent analysis (Figure 6of *Dolan et al., 2018*) of new whole brain EM data (*Zheng et al., 2018*) suggests that within one animal, LHNs of the same cell type can receive varied numbers of PN inputs that could well account for the observed response differences. This is also consistent with the recent observations of *Jeanne et al. (2018)* who found that LHNs originating from the same GAL4 line showed similar but not identical profiles of PN input as revealed by optogenetic mapping; however that study did not attempt to formalize cell type definitions and therefore did not conclude whether this variability was associated with distinct cell types or variations in the inputs within a cell type.

Finally, we would also like to emphasize that a requirement of co-clustering of LHNs by odor response alone is actually very stringent. PNs are generally assumed to be highly stereotyped odor responders, but we found that many PNs were not perfectly clustered using the approach of cutting a dendrogram formed by clustering odor responses. This was also true for earlier results of *Murthy et al. (2008)* (obtained with 7 cell types and 12 odors vs 22 cell types and 36 odors).

## Fine scale anatomical clustering confirms LHN classification

We next wanted to compare and cross-validate our manual classification into functional cell types with automated clustering. We were specifically concerned with the finest level of classification (i.e. cell type) and whether odor tuning differences (among cells with similar coarse anatomy) or fine anatomical differences would individually be sufficient to define a cell type. We selected all the functional cell types for which we had more then three filled, traced and co-registered cells (122 out of a total of 141; 42 cell types) and began by dividing them into six pools based on their primary neurite tract. We then carried out automated clustering based either on odor response profile (*Figure 6A*) or NBLAST clustering of neuronal morphology. Automated clustering of each of these pools reliably identified our manually defined physiology classes with a median Adjusted Rand Index for anatomy of 0.64 and 0.60 for odor response data (*Figure 7B*). This result demonstrates that the manual classification strategy in of the previous section is well-grounded. It also strongly supports the

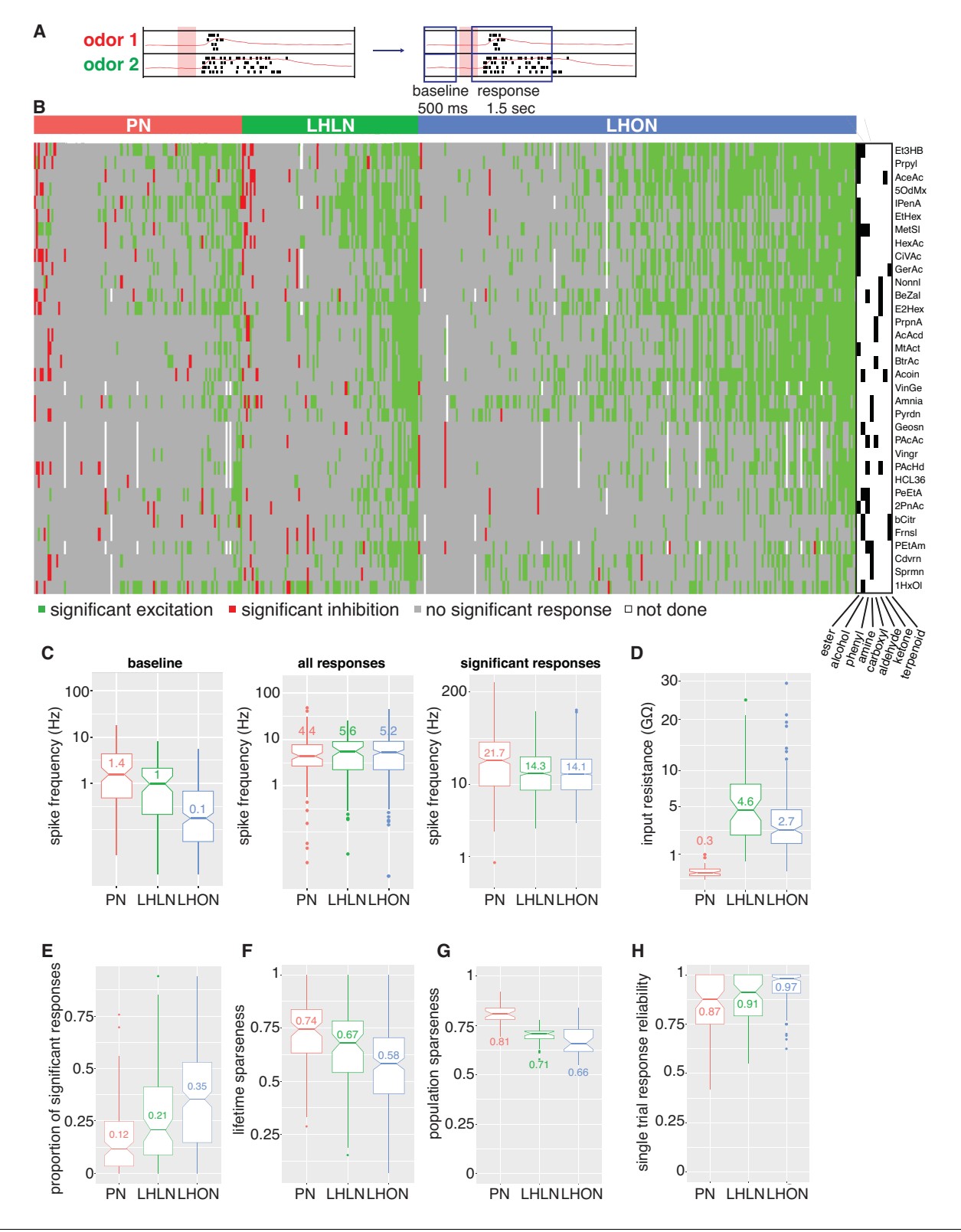

**Figure 4.** Population summaries of second and third order olfactory neurons. (**A**) Diagram of time windows used when identifying significant spiking responses (see Materials and methods). (**B**) Matrix showing significant spiking responses of PNs, LHLNs, and LHONs (colors match **Figure 3**) to different odors. A black and white matrix shows the chemical groups of the different odors (see supplementary spreadsheet). (**C**) Comparing firing rates of PNs, LHLNs, and LHONs. Baseline firing rate (baseline), firing rate in the response window (all responses), and firing rate in the response window for

*Figure 4 continued on next page*

*Figure 4 continued*

significant responses only (significant responses). LHONs have a lower baseline firing rate and, when using significant response only, a lower odor-evoked firing rate then PNs. (**D**) Input resistance of PNs, LHLNs, and LHONs. (**E**) Different measures of sparseness of odor responses in PNs, LHLNs, and LHONs showing that LHONs are broader then PNs. (**F**) Single trial response reliability using a threshold of 5 Hz for PNs, LHLNs, and LHONs. Responses were reliable for all three groups with LHON responses slightly more reliable probably due to differences in baseline firing rate.
DOI: https://doi.org/10.7554/eLife.44590.010

interpretation that our functional cell types are bona fide cell types since they can be independently defined by both anatomical and functional properties.

We also carried out the same cluster analysis across the whole dataset that is without dividing the neurons into six pools (black dots in *Figure 7B* to D). Although NBLAST anatomical clustering continued to perform well (Adjusted Rand Index, ARI = 0.74), hierarchical clustering of odor response data performed considerably worse (ARI = 0.38) albeit still above the chance level of ARI = 0 (*Figure 7B*). This lower performance results from the confusion of cells with rather similar response properties but very different morphology (*Figure 7—figure supplement 1B*).

In three cases, neither automated clustering by odor response or NBLAST clustering could reliably separate similar cell types defined during our manual classification. We carefully scrutinized the odor responses and morphology of the relevant cells (*Figure 7—figure supplement 1*. In two of these cases, we eventually decided to merge closely related cell types. In the third case, we concluded that our initial manual classification was correct. This resulted in a set of consensus cell types based on all the information at our disposal. We then reran our automated clustering across the whole dataset. While the performance of our anatomical clustering improved somewhat, the functional clustering continued to perform poorly (*Figure 7C*). For this set of consensus cell types we find the percent of correct classification by physiology or anatomy (*Figure 7D*) as 86% and 66%, respectively, when considering all cell types in a single pool. Finally we show a hierarchical NBLAST clustering for all cell types in *Figure 7E*, showing excellent agreement between the automated anatomical clustering and manually defined functional cell types.

In conclusion these results demonstrate the existence of 33 cell types in the LH with stereotyped odor responses and neuronal morphology across animals. They also strongly support the idea that the >165 LHN cell types that we have defined based on anatomical criteria alone will also show stereotyped odor responses across animals.

## LHONs sample odor space in a non homogeneous manner

The odor response cross-correlation heatmaps presented in *Figure 6B–D* are noticeably different for PNs, LHLNs, and LHONs. First, the mean correlation across cells is significantly higher for LHONs than for PNs or LHLNs. Second, the LHON heatmap shows considerable off-diagonal correlation structure that is largely absent from the PN and LHLN heatmaps. These two differences are obviously not independent – the higher overall correlation across LHONs may also result in more neurons with overlapping odor response profiles. It is important to understand the nature and origin of these differences between second order PNs and third order LHONs since the correlation structure of odor responses across each neuronal population will have a significant impact on its odor coding capacity.

One trivial explanation for the high cross-correlation between LHON responses in (*Figure 6B*) is that we are repeatedly sampling a small number of cell types many times. Having defined and validated LHN cell types in the previous two sections, we can set this trivial explanation aside by generating new heatmaps in which we aggregate all of the odor response data for recordings from the same cell types (*Figure 8*). For the LHONs in our dataset this resulted in a a 38 × 38 cell type heatmap. Comparing *Figure 8A* vs B-C, it should be clear that while there are PNs with strongly correlated odor responses, these groups are located along the diagonal. In contrast for the LHON heatmap, there are many squares with high correlation far from the diagonal: many cell types have odor responses that are correlated with multiple groups of cell types.

We considered three possible explanations for the differences that we see between PN and LHON heatmaps.

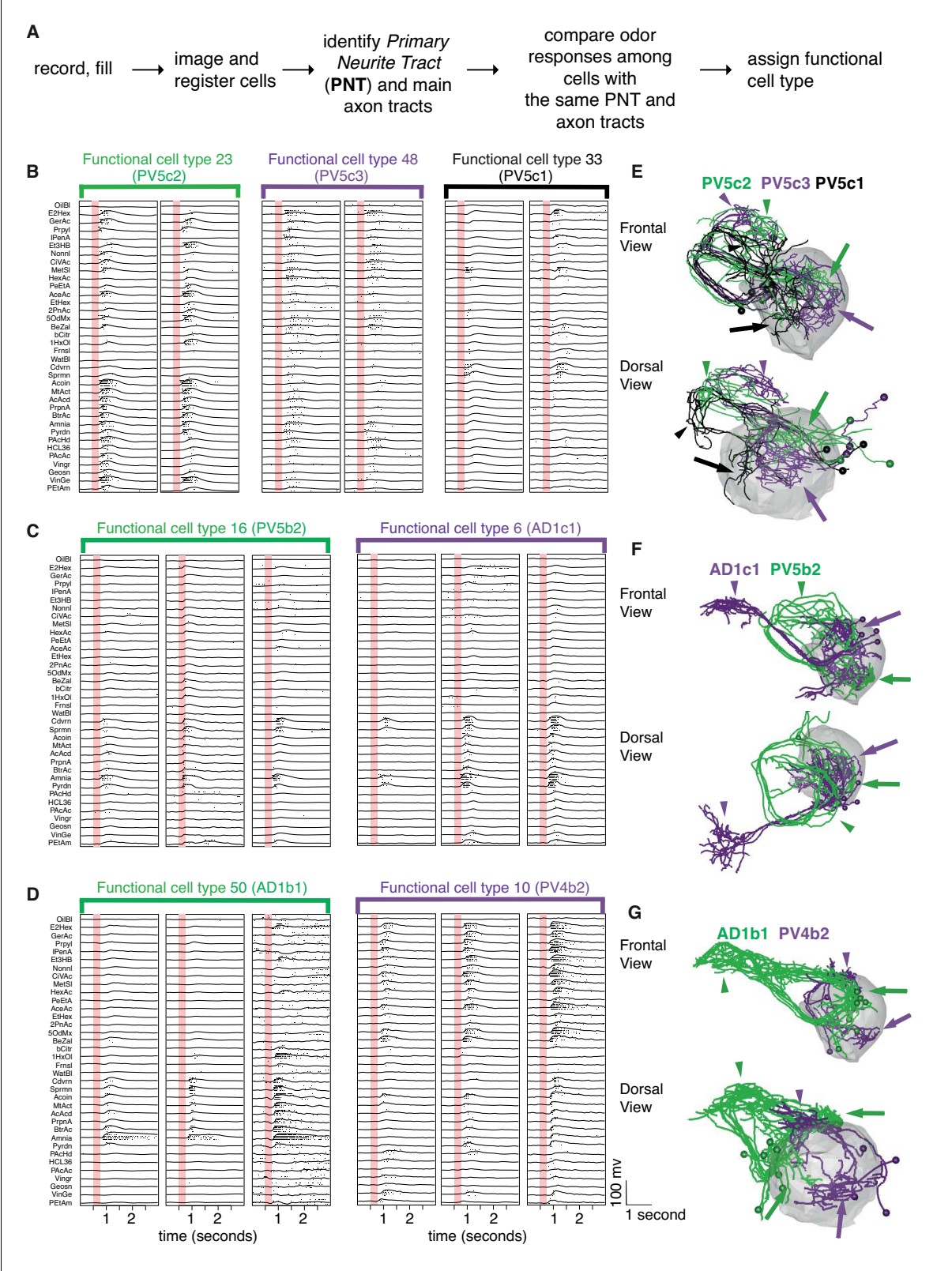

**Figure 5.** Comparing physiology and anatomy of different cell types. (A) Our pipeline for identifying cell types. (B) Odor responses for three cell types (n = 2 neurons each) belonging to the same anatomy group. Each pair of neurons that belongs to the same functional cell type are marked by a colored bar with the functional cell type number. Although the three cell types have really similar anatomy, their responses are clearly very different. (C) Example of two functional cell types (n = 3 neurons each) that have very similar response properties but very different anatomy. (D) Some cell types

*Figure 5 continued on next page*

*Figure 5 continued*

showed significant variability in the strength of their odor responses across recorded cells while still showing consistency (n = 3 neurons for each of 2 cell types). (E, F, G) Frontal (top) and dorsal (bottom) images of dye filled neurons correspond to the functional cell types in B-D. Colored arrows (dendritic) and arrowheads (axonal) pointing to the arbors of each cell type. Altogether the figure demonstrates cons functional cell type classification.
DOI: https://doi.org/10.7554/eLife.44590.011

Tuning Breadth - LHONs are more broadly tuned, responding to significantly more odors than PNs. More broadly tuned cells may be more correlated with other cells, potentially explaining the increased cross-correlation across cell types.

Cell Numbers - There are more LHONs then PNs. As the number of cells and cell types sampling the same odor space increases, differences between cell types are bound to decrease resulting in increased cross-correlation across cell types.

Odor Space - LHONs sample odor space in a biased manner, with many cell types responding to related sets of odors, leading to higher cross-correlation.

To try to distinguish these different possibilities, we first carried out a simple computational experiment in which we shuffled the odor stimulus labels, choosing a different permutation for each cell type. This procedure maintains the same firing rates and tuning breadths for each cell type, but disrupts the correlation structure across cell types that would result from extensive similarities in odor profiles across types. *Figure 8A'–C'* presents the results of this manipulation, which clearly removes the off-diagonal structure for all three groups of cells, leaving almost no cases of elevated cross-correlation. This effect is clearly much larger for LHONs than for PNs (quantified in *Figure 8D*). Increased tuning breadth alone therefore cannot explain the extensive off-diagonal cross-correlation structure for LHONs, but instead biases in the odor response properties across cell types appear to be the main factor.

This first analysis suggests that in the absence of strong biases in the odors that excite LHONs, tuning breadth has no substantial impact on the population cross-correlation. However, given that LHONs have such response biases, we next examined whether tuning breadth could be a contributory factor. We found a statistically significant (p=0.004) although not particularly strong (adjusted $R^2$ = 0.19) positive relationship between mean odor response probability and mean cross-correlation scores. However we can also see that if we compare with PNs, LHONs had consistently higher cross-correlation scores, even for cell types with low odor response probabilities (left of red line in *Figure 8E*). One issue with this last comparison is that the mean cross-correlation scores for narrowly tuned LHONs still included comparisons against both broad and sparse LHONs. We therefore further limited our analysis to consider the cross-correlation only between sparse LHON cell types. Once again LHONs showed higher cross-correlation scores (p=2.7E-5) than PNs (*Figure 8F*). This also indicates that the number of cell classes is not the main reason for the high correlation since by limiting our analysis to sparse classes only we also matched the number of PN and LHON classes (20 and 22 classes, respectively).

Summarizing, we conclude that LHONs sample odor space less homogeneously than PNs leading to higher cross-correlation in LHON responses than their PN inputs. We further show that increased tuning breadth of LHONs and the increased number of LHON classes are not the main reason for this high correlation.

## Encoding of odor categories

We have already explored a number of aspects of odor coding by LHNs. For example we have seen that LHNs, as a population, respond to 3 times more odors than their PN inputs and that they sample odor space inhomogeneously. We hypothesized that these features of LHN odor coding arise because they pool specific odor input channels that signify odors of common behavioral significance. The circuit origins of behavioral significance, which can be summarized at its very simplest level as a binary valence – whether odors are attractive or repulsive – have received considerable attention recently (reviewed by *Knaden and Hansson, 2014*). However the observed behavioral valence is extremely dependent on numerous factors including the exact behavioral paradigm and odor concentration used. Therefore rather than trying to examine LH odor coding from the perspective of the

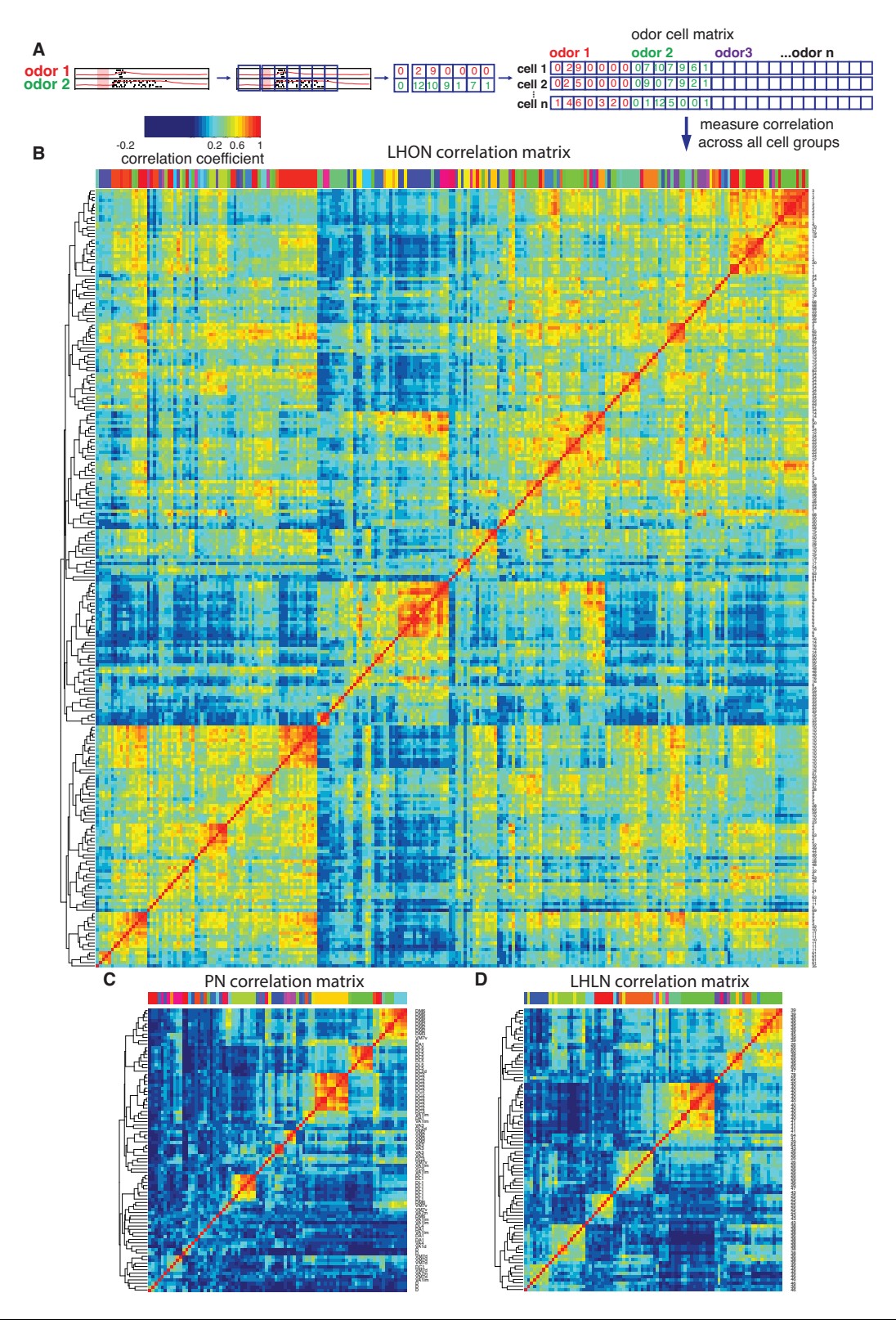

**Figure 6.** Cross-correlation clustering of odor response data for PNs, LHLNs, and LHONs. (**A**) Analysis pipeline for generating the cell-odor response correlation matrix and measuring correlation across cells. Responses were binned (blue squares, 50 ms) and the mean firing rate was calculated for each time bin. For each cell, the responses to all odor, ware concatenated into a single vector and a matrix of all the cell odor responses was generated. This cell-odor matrix was used to calculate the Pearson's correlation between the odor responses for all pairs of cells. (**B–D**) Heatmaps of the resultant cross-

*Figure 6 continued on next page*

*Figure 6 continued*

correlation matrices for LHONs, PNs and LHLNs, respectively. To allow comparison all three heatmaps share the same color scale for the correlation coefficient (top left) demonstrating a higher correlation between LHONs as well as considerable higher level of off-diagonal correlation structure.

DOI: https://doi.org/10.7554/eLife.44590.012

behavioral valence reported for different odors in the literature, we initially focussed on encoding of well-defined chemical features.

We first categorized our odor set based on the presence of alcohol, aldehyde, amine, carboxyl, ester, phenyl chemical groups. We then examined odor encoding at the population level of PNs, LHLNs, and LHONs using principal components analysis. The first principal component consistently encoded response magnitude (data not shown). *Figure 9A* shows the population response trajectories projected into the space of the second and third principal components, and color-coded by odor category. Two features of this analysis seemed particularly noteworthy. First, there was a progressive increase from PNs to LHONs in how spread out odor representations were in this principal component space. Second, LHON responses appeared to separate certain odor categories, especially amine containing odors (typical of decomposing biological matter) versus acetates (typically light, fruity odors).

This result motivated us to examine the ability of individual LHNs to encode odor categories. We treated each cell as a binary classifier for a given odor category, that is signaling the presence or absence of that category and measured its performance using a normalized area under the ROC curve (AUC) score (see Materials and methods). LHONs and, to a lesser extent, LHLNs, but not PNs convey category information in their odor responses, when compared with shuffled control distributions (*Figure 9B*). The LHON population has the largest fraction (70%) of category-informative cells, followed closely by LHLNs, which have nearly twice as many category-informative cells as PNs (data not shown). Among the six categories, four were highly represented in the LHON population and amine categorizers appeared to be the most selective (*Figure 9C*). These results indicate that LHNs indeed develop a novel ability to encode higher order odor features that are more likely to be behaviorally relevant to the fly, confirming a longstanding hypothesis in the field.

As noted earlier, PCA analysis suggested that population odor responses were increasingly spread out moving from PN through LHLN to LHON population responses. In the case of PNs, the second and third principal components that are plotted in *Figure 9A* are dominated by a small number of odors with large values. We wondered if this might conceal a more regular structure as we observed for LHONs. We therefore repeated the PCA analysis after carrying out an adaptive normalization procedure boosting responses to all odors (see Materials and methods). However as shown in *Figure 9D*, this did not reveal any strong categorical organization in PNs.

We also asked whether there was any association between particular odor categories and brain regions within and outside the lateral horn (i.e. locations of LHON dendrites and axons, respectively). We used a clustering approach to partition these regions into 25 compact 'supervoxels' (see Materials and methods) and then asked if these were associated with LHONs encoding particular odor categories. Both within the LH and in the axon target regions in the superior protocerebrum we found regions that were strongly associated with four of the odor categories (*Figure 9E*, amines, esters, aldehydes and carboxylic acids. These included several well-separated domains in the output regions, suggesting that there might be distinct groups of downstream target for LHONs with different category specificity.

Given the improved selectivity of individual LHONs for specific odor categories we also tested the ability of populations of LHNs to identify odor category or identity. To do this we repeatedly generate random subpopulations of cells for a given number of cell types, where each cell was the sole representative of a particular cell type. We then trained linear support vector classifiers (SVC) to perform either identity or category decoding on a trial-by-trial basis for each time bin (see Materials and methods), selecting a single SVC tuning parameter that maximized accuracy using one half of the random samples. We then reported performance summaries for the other half of the random samples using the selected SVC tuning parameters.

As we expected, LHN populations were better than PNs and LHLN at identifying odor category (*Figure 9F*). At the very low end of the graph (just one cell type used for decoding) only LHONs

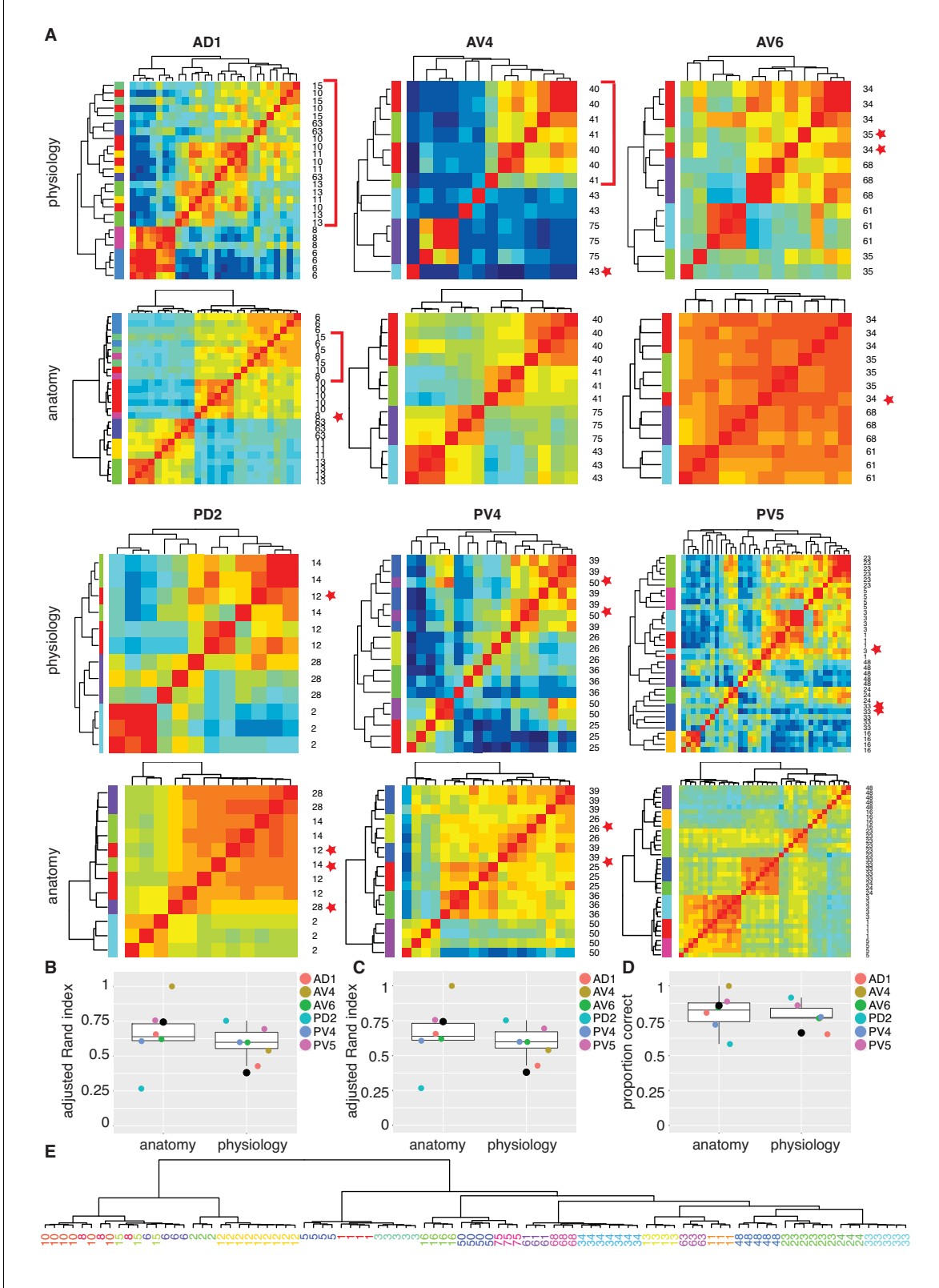

**Figure 7.** Comparing anatomy and physiology classification. (**A**) Cross correlation matrix of odor responses and fine anatomy (NBLAST) for the same cells. Cells were divided according to their PNT. Only classes with at least three traced cells were used. We highlighted areas of misclassification with either a star (single mis-classified cell) or a red bar for a section of several cells. Correlation matrices for six primary neurites were organized in pairs with physiology on top and Anatomy below. Color scale for all physiology Correlation matrices and all anatomy Correlation matrices is the same. (**B**)

*Figure 7 continued on next page*

*Figure 7 continued*
Summary comparing Adjusted Rand Index clustering score for each primary neurite tract by physiology and anatomy. Black dot in B to D marks the results of analyzing the entire data set together. (C) Summary comparing Adjusted Rand Index clustering score for each primary neurite tract by physiology and anatomy after correcting class labels in two cases. (D) Summary comparing percent correct clustering score for each primary neurite tract by physiology and anatomy after class correction. (E) NBLAST clustering of all functional cell types with >=3 traced cells after merging two cases of indistinct cell types (see *Figure 7—figure supplement 1* for details). Note the excellent agreement between the anatomical clustering and our manually defined functional cell types.
DOI: https://doi.org/10.7554/eLife.44590.013
The following figure supplement is available for figure 7:

**Figure supplement 1.** Morphologically and Physiologically Similar Classes.
DOI: https://doi.org/10.7554/eLife.44590.014

could identify odor category when compared with shuffled control distributions, consistent with the ROC analysis in *Figure 9B*. They maintained their superior performance over the entire range number of cell types tested. One important question for future work would be to determine what is the biologically relevant size of LHON population that might feed into downstream neurons – the anatomy of LHONs with axon projections to disparate higher brain areas make it unlikely that there would be integration of large numbers of LHON cell types.

In comparison, neither PNs, LHONs, or LHLNs showed above chance odor identification performance when only one class was used (*Figure 9G*). As we increased the number of cell types in the decoding population, performance improved for all groups but LHONs gradually improved compared with PNs and LHLNs. In future it will be interesting to compare these results with KCs by obtaining odor response data in equivalent conditions. While this analysis is certainly consistent with our hypothesis that stereotyped integration in the LH could enable genetically determined categorical odor representations, there is also a weakness that cannot obviously be overcome without additional data: this population decoding analysis tests the ability of random subsets of neurons to predict the category or identity of all our test odors. However given the stereotyped nature of LHN responses, we propose that particular LHN populations are dedicated for particular odor categories. In this regard, future analysis based on comparing the responses of LHNs integrating known olfactory channels will be instructive.

## Integration of odor channels by LHNs

Many of our observations (increased tuning breadth, increased single cell categorization ability, reduced representational dimensionality) suggest that the LHONs pool olfactory information to better inform the behavioral significance of an odor. Although this provides a rationale for the observed differences in odor coding, it does not account for them mechanistically. Previous light level studies have attempted to predict PN to LHN connectivity (*Jefferis et al., 2007*) based on light level overlap and a handful of these predictions have been validated (or refuted) electrophysiologically (*Fişek and Wilson, 2014*). Nevertheless light level mapping, especially when carried out across brains cannot reliably predict actual synaptic connections.

In order to compare our observations about LHN odor coding with measurements of the PN to LHN convergence ratio, we leveraged a newly available whole brain EM dataset (*Zheng et al., 2018*). This allowed us to obtain direct information about how LHN dendrites integrate inputs from different PN axons. We selected an anatomically diverse sample of 29 LHONs and 17 LHLNs derived from 10 and 5 primary neurite tracts, respectively (seven neurons are from *Dolan et al., 2018*, the remainder P Schlegel, ASB, GSXEJ, in preparation). These neurons were reconstructed to completion in the lateral horn, enabling us to analyze the complete repertoire of excitatory PN input onto their dendrites originating from 51 glomeruli (see Materials and methods).

All neurons analyzed had a small number of strong inputs but most had a long tail of weaker inputs (*Figure 10A*). Therefore although they received at least one input from 13 glomeruli on average (range 0–33), if we considered only those glomeruli accounting for more than 1.5% of total input synapses to a given LHN, then LHLNs received 5.0 ± 1.8 significant inputs from uniglomerularnputs from uniglomerular excitatory olfactory PNs, and LHONs received 5.2 ± 2.9 (mean Â ± standard deviation; *Figure 10B*) with a range of 0–13. One AD1b1 LHON received no uniglomerular excitatory olfactory PN input, despite having dendritic innervation in the LH, indicating that some LHN types

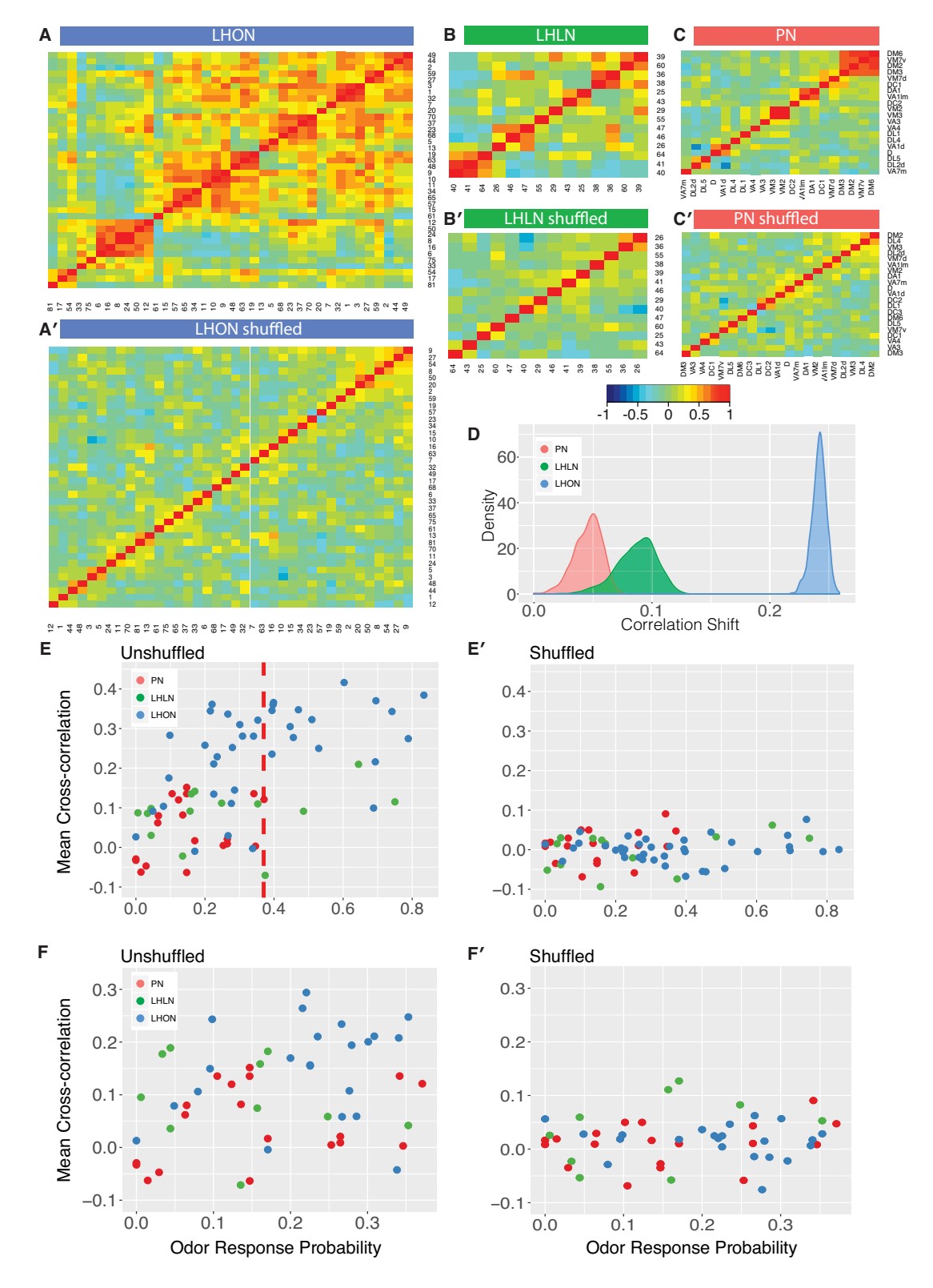

**Figure 8.** Comparing odor coding of LHNs with their inputs. (A–C) Aggregated correlation heatmaps for LHONs, LHLNs, and PNs generated by calculating a mean odor response profile for each cell type and then computing the correlation matrix across all cell types. (A'–C') Aggregated correlation heatmaps calculated after shuffling odor labels. For comparison all six heatmaps share the same color scale. (D) Histogram of mean correlation shift by randomizing the odor labels (n = 1000 replicates). (E, E') For each cell type the mean cross-correlation against all other cell types

*Figure 8 continued on next page*

*Figure 8 continued*

was plotted against the proportion of significant odor responses of that cell type, either with or without shuffling of odor labels. (**F, F'**) As for E but only including correlation between sparse cell types (left to the red line in E indicating p(response)¡0.36, the highest odor response probability for PNs). Note that F and F' are not just a subset of E and E' as we recalculated the mean cross-correlation after selecting the sparsest LHON and LHLN cell types. Altogether the figure shows that the higher cross-correlation in LHON responses is due to LHONs sampling of odor space less homogeneously than PNs and not because of increased tuning breadth of LHONs or the increased number of LHON classes.

DOI: https://doi.org/10.7554/eLife.44590.015

receive majority input from other neurons within the LH. These numbers provide a key parameter to start modeling the circuit origins of the odor coding properties of LHNs (*Litwin-Kumar et al., 2017*). They are also comparable with recent optogenetic mapping observations on PN-LHN functional connectivity which obtained an estimate of 4.8 glomeruli/LHN based on optogenetic mapping of PNs innervating 39 glomeruli (*Jeanne et al., 2018*).

We also compared these LHN input numbers with those for MB Kenyon cells, re-analyzing prior light level (*Caron et al., 2013*) as well as new EM data (*Zheng et al., 2018*). The cell type weighted averages for glomerular inputs onto Kenyon cells were comparable across both data sources (EM, $5.2 \pm 2.9$, LM, $6.0 \pm 1.2$), although the EM average was slightly lower, possibly due to differences in how the two studies counted claws. Intriguingly the mean number of glomerular inputs is highly comparable between KCs and LHNs despite large differences in odor coding properties. However there are obvious differences: Comparing EM data, there are many fewer synapses onto KC dendrites compared with LHONs ($77 \pm 29$ vs $467 \pm 278$). Secondly the proportion of feed-forward excitatory PN inputs was much lower for LHONs ($79 \pm 5\%$ vs $41 \pm 14\%$). In both cases the majority of other inputs appear local in origin, with the single giant APL inhibitory neuron being the major remaining input for KCs (*Papadopoulou et al., 2011*; *Zheng et al., 2018*), while LHONs received many distinct inputs as we have recently shown in *Dolan et al. (2018)*. In the future it will be extremely interesting to determine how intrinsic and local circuit properties shape the very different odor coding by neurons in the LH and MB.

## Discussion

### Odor coding in LH

Our principal finding is that lateral horn neurons (LHNs) as a population are genetically and anatomically defined cell types with stereotyped odor responses. Starting from recordings of genetically defined populations we cross-validated fine scale anatomical differences and odor tuning for 37 LHN cell types; this confirms that stereotypy is a general feature of the lateral horn (LH) and not particular to specialist odor pathways such as those that process pheromone information, which may retain a labeled line logic all the way from the periphery. Although we see evidence of narrowly tuned LHNs dedicated to the processing of specific odors, the population as a whole shows 3x more odor responses than their olfactory projection neuron (PN) inputs. The increased tuning breadth may reflect a transition to a more behaviorally relevant coding scheme. This is consistent with our findings that LHNs show significantly improved odor categorization compared with PNs, apparently due to stereotyped pooling of related odor channels. The chemical categories that we analyzed are probably not of direct ethological relevance to the fly, but serve as proxies – further explorations of olfactory neuroecology are clearly necessary. For example we saw limited evidence for simple representations of olfactory valence in LHN responses.

It is instructive to compare the odor tuning properties we find across the lateral horn with those reported for the *Drosophila* mushroom body. Major differences in the mushroom body (MB) include the lack of response stereotypy (*Murthy et al., 2008*) and sparser odor tuning (*Turner et al., 2008*); the distribution of odor tuning in the LH also appears to be wider – that is LHNs appear more functionally heterogeneous. However, there are also similarities – there is divergence of PNs onto a larger population of third order neurons in both cases. Furthermore baseline firing rates are very low in both LHNs and Kenyon cells (KCs) and the evoked firing rates are also lower than in their PN input. This could reflect energetic, spike economy considerations or a need to binarize neural responses prior to memory formation or organizing behaviors.

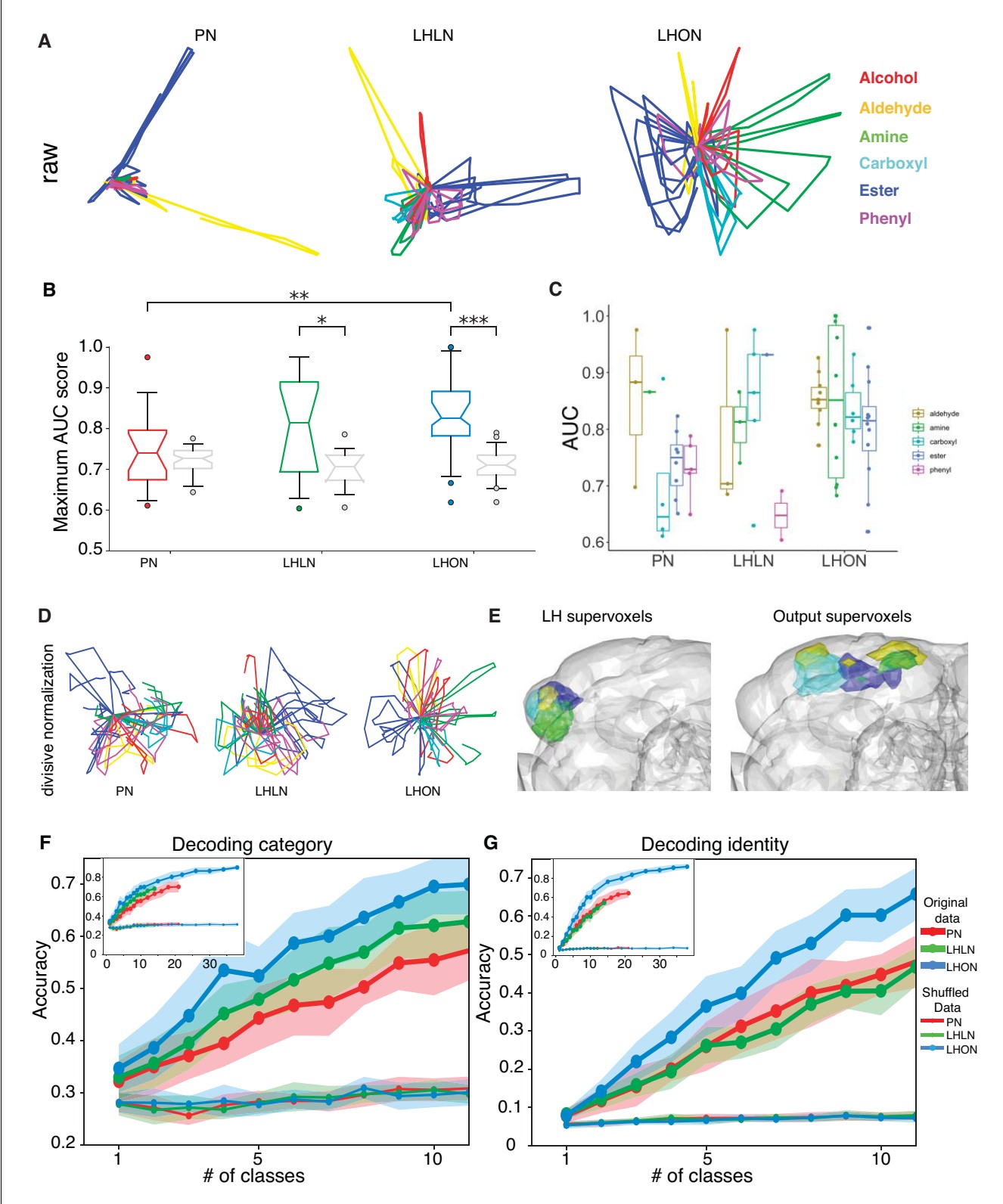

**Figure 9.** Odor categorization in the LH. (**A**) Population representations of odors. Responses are projected into the spaces of the second and third principal components, and color-coded by odor category. (**B**) Distribution of AUC scores for PNs and LHLNs, and LHONs. An AUC score of 0.5 indicates no information about odor category. Box = 25–75% centiles, line = median, whiskers 5–95% centiles, notch indicates bootstrap 95% confidence interval of the median. LHON odor responses convey more category information than PNs ($p<0.005$), one-sided Mann-Whitney U-test (**C**)

*Figure 9 continued on next page*

*Figure 9 continued*

Distribution of AUC scores for each population divided into the different odor categories (D) PCA analysis after divisive normalization. (E) Mapping of the different odor categories to brain voxels (see Materials and methods: Odor coding analysis). (F–G) Decoding accuracy of linear support vector classifiers (SVC) trained to perform category (F) or identity (G) classification using different numbers of cell classes. The main figure shows the result of using 1 to 10 classes while the inset shows the result of using all available classes for each group. Altogether we show that LHONs are better at encoding odor chemical categories than PNs.

DOI: https://doi.org/10.7554/eLife.44590.016

It is also interesting to compare response properties with recent recordings from the mammalian posterolateral cortical amygdala (*Lurilli and Datta, 2017*), which has been compared to the LH, since it receives spatially stereotyped input from the olfactory bulb (*Sosulski et al., 2011*) and is required for innate olfactory behaviors (*Root et al., 2014*). (*Lurilli and Datta, 2017*) found that odor tuning properties were very similar to the mammalian piriform cortex (which has been compared to the mushroom body). Both regions showed decorrelated odor representations (whereas we find that LHN odor responses show significant correlations suggestive of a focus on particular combinations of olfactory channels) and odor tuning in the cortical amygdala was actually somewhat sparser. In further contrast to our observations in the LH they found no evidence for categorization of odors by chemical class and crucially no evidence for response stereotypy in a way suggestive of stereotyped integration of defined odor channels. We would however caution with respect to the last point that

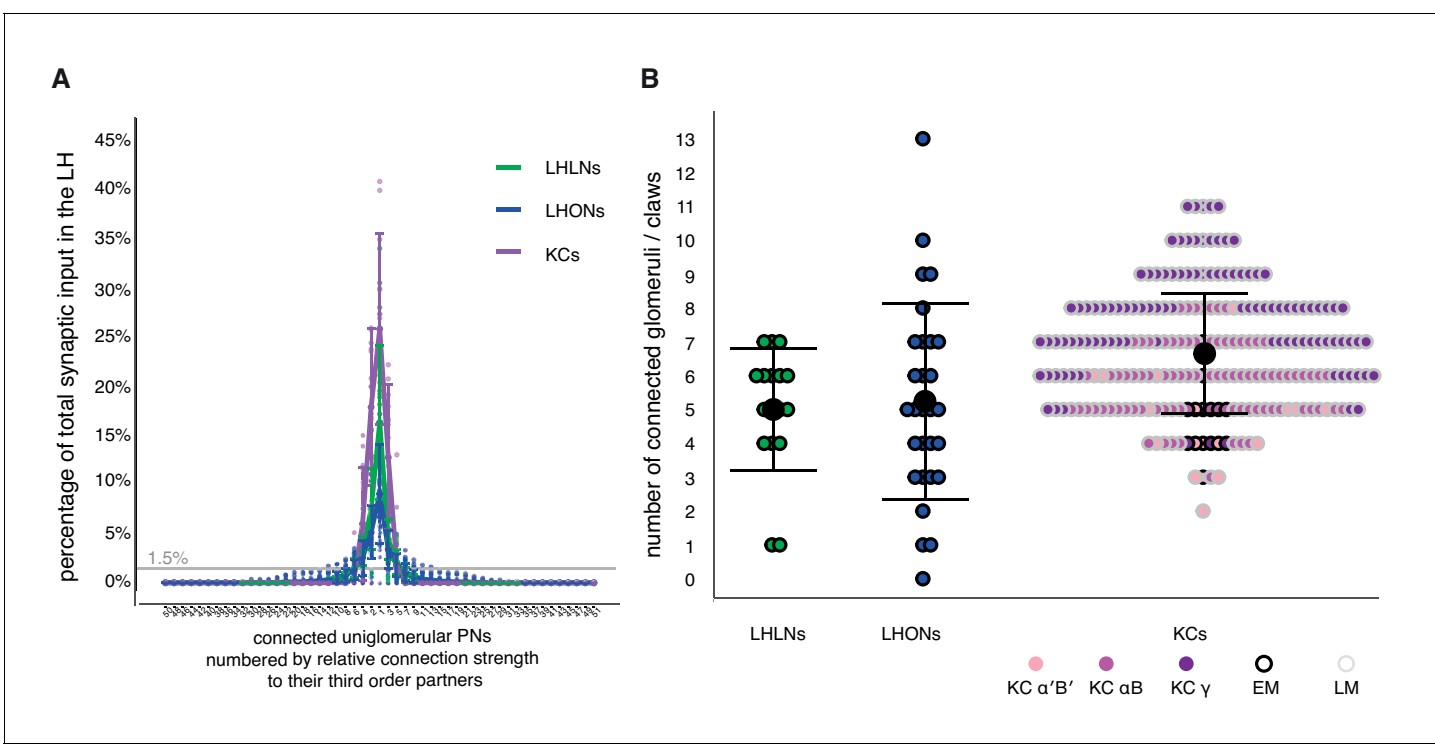

**Figure 10.** Structural connectivity between antennal lobe glomeruli and third order olfactory neurons. (A) Input tuning curves for a set of EM reconstructions comprising 17 LHLNs, and 29 LHONs (Schlegel, Bates et al. *ms in prep*) whose dendrites are restricted to the LH. The x axis is rank ordered by the number of synaptic inputs from uniglomerular PNs from each of 51 glomeruli. (B) Dot plots showing distribution of number of input glomeruli for LHLNs, and LHONs, and the number of dendritic claws for KCs. KC claw number data was based on light microscopy (LM) data presented in *Caron et al. (2013)* or EM reconstructions in *Zheng et al. (2018)*. Some LHNs receive no excitatory uniglomerular PN input; their feed-forward input may come from multiglomerular or non-olfactory projections to the LH. Group means are shown by large black dots, error bars indicate a single standard deviation from the mean. See also Materials and methods. In sum, a reasonable cut-off for the observed structural connectivity in our EM-reconstructed wiring diagram reveals that the average LHN and each KC may sample the same number of glomeruli, via uniglomerular excitatory olfactory PNs. LHNs exhibit a larger standard deviation; some LHNs may act in a specific labelled line for unique and behaviorally significant odors, others may need to sample a larger number of ethologically connected odors more broadly.

DOI: https://doi.org/10.7554/eLife.44590.017

had we recorded from a small fraction of randomly selected neurons of the *Drosophila* LH, we might easily have missed response stereotypy. It is only because we were able to use genetics to bias our sampling, and also to record from a significant fraction of the whole LH population, that we could obtain clear evidence for odor response stereotypy. Nevertheless, these differences seem marked and it will be very interesting to compare the logic of these systems across organisms. One point to note is that the circuits in the fly may be more compact: LHNs can in a few cases connect directly to fourth order neurons with descending projections to the nerve cord likely to have a direct impact on motor behavior (*Ruta et al., 2010*; *Huoviala et al., 2018*).

## Circuit mechanisms

There are some similarities between the increase in tuning breadth that we observe at the PN-LHN transition and what has previously been reported at the first synaptic layer of the olfactory system (the olfactory receptor neuron to PN synapse). In the antennal lobe broadening appears to depend on a compressive non-linearity, which boosts weaker inputs (*Bhandawat et al., 2007*) and possible excitatory local interactions (*Olsen et al., 2007*; *Shang et al., 2007*). Although a direct comparison between the extent of broadening in the antennal lobe and LH is not possible without measuring odor responses from many receptor neurons under the same stimulus conditions (as we did for PNs and LHNs) it seems likely that the effect is larger in the LH. Importantly the mechanism here appears quite different, with direct pooling of feed-forward inputs.

Our initial EM connectomics observations suggest that a typical output LHN (LHON) receives strong inputs from 3 to 7 excitatory PNs albeit with a long tail of weaker connections, some of which are likely to have an impact. Intriguingly this number (referred to as the synaptic degree, K, *Litwin-Kumar et al., 2017*) is not that different from the seven inputs reported for KCs in the mushroom body (*Caron et al., 2013*). How is it that LHONs and KCs listen to rather similar information but produce very different responses? It is true that the inputs received by LHNs will in general be more highly correlated; this is both because LHNs appear to receive input from all the PNs originating from a given glomerulus (when there are >1) and because those PNs coming from different glomeruli often have related odor tuning (*Dolan et al., 2018*; *Jeanne et al., 2018*). Nevertheless, we argue strongly that the rules of integration that result in broadening in LHONs and a sharp reduction in tuning breadth in KCs are likely to differ significantly. *Fişek and Wilson (2014)* have shown that LHON firing rates scale linearly with their PN inputs, while *Gruntman and Turner (2013)* showed that KC membrane potential linearly integrates dendritic inputs. Differences in the integrative properties could result from both intrinsic and circuit mechanisms (i.e. local interneuron interactions), but two factors likely to have a major impact are the spatial distribution of synapses and the spike threshold. PN inputs are broadly distributed across LHON dendrites (Schlegel, Bates et al, in preparation), whereas PN inputs onto KCs are highly clustered at individual dendritic claws. The many individual connections at each KC claw may be integrated to produce a reliable response that is nevertheless usually below the spike threshold – therefore multiple input PNs must be co-active and KCs act as coincidence detectors. In contrast the inputs on LHON dendrites may be integrated in a more graded fashion with a lower spike threshold (*Fişek and Wilson, 2014*). Of course the biggest difference is that LHNs receive stereotyped inputs according to their anatomical/genetic identity (see *Dolan et al., 2018*) and this provides a mechanism for the odor response stereotypy that we observe.

We would also like to highlight some additional differences in circuit architecture between the MB and LH that may be of functional significance. First the MB calyx receives only excitatory PN input, whereas, there is a population of almost 100 inhibitory PNs that project to the LH (*Tanaka et al., 2012a*). Second we find that the LH contains an estimated 580 local neurons (most of which are inhibitory, *Dolan et al., 2019*), whereas the mushroom contains just one local inhibitory neuron, the APL. We suspect that a major reason for this difference is again related to the stereotyped vs non-stereotyped design of these two centers. The APL is not selective but appears to pool all KC inputs to implement a winner take all gain control mechanism, suppressing more weakly activated KCs (*Papadopoulou et al., 2011*). Our preliminary EM results show that at least some LHLNs integrate small numbers of input channels (2–3 strong inputs). We suggest that they then make stereotyped connections either reciprocally onto their input PNs or onto other specific neurons in the LH.

## Cell types in the central brain

There is renewed interest in the identification of cell types in the brain as an important step in the process of characterizing circuits and behavior (*Zeng and Sanes, 2017*; *Bates et al., 2019a*). Historically, cell types have been best classified by morphology and the most detailed work has been in the sensory periphery (e.g. 55 cell types in the mouse retina: *Masland, 2001*). Recently single cell transcriptomics has begun to match this morphological classification (*Shekhar et al., 2016*) and also to enable more detailed exploration of diversity in deeper brain regions (e.g. 133 cell types in mammalian cortex: *Tasic et al., 2018*). However, relating cell types to functional and network properties especially in higher brain areas remains challenging.

One of the major surprises from our work is our identification of 165 anatomically distinct LHN cell types; our cross-validation of anatomical and odor response properties for 37 leads us to believe that most of these will turn out to be functionally stereotyped as well. Furthermore our light level survey is incomplete; we predict that complete EM data could reveal more than 250 LHN cell types. In short there are more cell types in the lateral horn than have yet been identified in the whole of the mammalian neocortex (*Tasic et al., 2018*). This disparity raises a number of issues.

One interesting observation is that it was easier to identify cell types anatomically than by odor response profile alone. It has recently proven possible to characterize 30 retinal ganglion cell types in the mouse based solely on their visual response properties (*Baden et al., 2016*). It may be that this highlights a difference between the richness of achievable visual stimulation protocols with odor delivery; although our core 36 odor set was large by the standards of the field, this is still a small fraction of the world of possible odors for the fly. Nevertheless there appear to be many more LHNs than retinal ganglion cell types and we find examples of neurons that appear to be solely distinguished by their projection patterns (presumably defining different downstream partners) which are only revealed through anatomical characterization. For these reasons we believe that response properties alone are insufficient to define cell type and this seems likely to be the case in other higher brain areas.

Initial evidence from EM connectomics (*Dolan et al., 2018*) has shown that two specific LHN cell types integrate stereotyped sets of olfactory channels with similar odor response profiles. This is paralleled by the recent work of *Jeanne et al. (2018)*, who showed that morphologically similar neurons sampled from the same or different GAL4 lines showed similar functional connectivity; furthermore they showed that the patterns of co-integration were not random, but that certain pairings of PN inputs were over-represented in the PN population. These observations are likely to be at the heart of the category selectivity that we observe in LHON responses. It will be exciting to integrate functional and anatomical properties more deeply with circuit properties. Furthermore our genetic screening identifies at least 69 molecular profiles based on expression of driver lines (*Figure 1E*). This molecular diversity underlies our ability to generate cell type specific split-GAL4 lines in *Dolan et al. (2019)*. The existence of such a rich and coupled genetic and anatomical diversity raises interesting questions about how connection specificity can be achieved during development in this integrative brain area.

## What is the behavioral function of the lateral horn?

The lateral horn is one of two major olfactory centers in the fly. The hypothesis that it might play a specific role in unlearned olfactory behaviors dates back at least to *Heimbeck et al. (2001)*. This has been strengthened by observations about the relative anatomical stereotypy of input projections to the mushroom body and lateral horn (*Marin et al., 2002*; *Wong et al., 2002*; *Tanaka et al., 2004*; *Jefferis et al., 2007*; *Caron et al., 2013*). Nevertheless in spite of this general model of a division of labour between LH and MB, functional evidence has been hard to come by. Some arguments about LH function have been based on experiments that manipulate mushroom body neurons; here it is worth noting that there are olfactory projections neurons that target areas outside of these two principal centers (e.g. *Tanaka et al., 2012a*; *Aso et al., 2014b*) so the lateral horn cannot rigorously be concluded to mediate behaviors for which the mushroom body appears dispensable.

In this experimental vacuum a large number of hypotheses have been proposed for LH function. One obvious suggestion based on anatomy was that LHNs should integrate across olfactory channels (*Marin et al., 2002*; *Wong et al., 2002*). Of course integration can have opposing effects on tuning. For example (*Luo et al., 2010*) proposed that LHNs might have highly selective odor responses and

early recordings from narrowly tuned pheromone responsive neurons are consistent with this idea (*Ruta et al., 2010*; *Kohl et al., 2013*). However *Kohl et al. (2013)* also observed more broadly tuned neurons that clearly integrated across olfactory channels and *Fişek and Wilson (2014)* showed quite linear integration of two identified olfactory channels. Our electrophysiological recordings together with first EM connectomics results suggest that integration across multiple odor channels and broadening of odor responses are the norm.

Turning to the biological significance of LHNs for the fly, one suggestion, based on anatomically discrete domains for food and pheromone odors, is that the LH might organize odors by behavioral significance (*Jefferis et al., 2007*). Others have suggested that the LH might mediate innate responses to repulsive odors only (*Wang et al., 2003*) or that the LH might organize odor information by hedonic valence (*Strutz et al., 2014*). Although our survey of LHN odor responses is not yet conclusive on any of these points, we did find clear evidence for an improved ability to categorize chemical groups of odorants (*Figure 9*). Further work integrating more information about the behavioral significance of different odors should be instructive.

One synthesis of these different ideas is that the MB performs odor identification, whereas the LH/lateral protocerebrum performs odor evaluation, both learned and innate (*Galizia, 2014*). Although we have no evidence to support a direct role for the LH in evaluation of learned olfactory signals, new work from our group has identified a class of lateral horn neurons that integrates both innate (directly from the antennal lobe) and learned olfactory information (from MB output neurons) of specific valence; these LHNs are required for innate appetitive behavior as well as learned aversive recall (*Dolan et al., 2018*). We have also identified multiple LHN axon terminals as targets of mushroom body output neurons, suggesting that mushroom body modulation of innate olfactory pathways may be a general strategy of learned behavioral recall (*Dolan et al., 2018*; *Dolan et al., 2019*). These results emphasize the extensive interconnection between these brain areas and should caution against oversimplifying their distinct roles in olfactory behavior. Nevertheless synthesizing the results in this study with other new work (*Dolan et al., 2018*; *Dolan et al., 2019*; *Huoviala et al., 2018*; *Jeanne et al., 2018*) does support the hypothesis that stereotyped integration in the LH could support genetically determined categorical odor representations, while the MB may enable identification of specific learned odors.

We finally return to a key question posed at the start of the manuscript: why does the LH need so many cells and cell types? At this stage we would suggest that LHNs are likely to show both stereotyped selectivity for odor categories and specificity for different aspects of odor-guided behavior. Specific combinations of the same odor information could be used to regulate distinct behaviors by targeting different premotor circuits (see *Figure 5*). Indeed our group has recently identified a requirement of a specific LHN cell type (AV1a1) in egg-laying aversion (*Huoviala et al., 2018*) to the toxic mold odorant geosmin (*Stensmyr et al., 2012*) even though this is one of more than 70 cell types that receive geosmin information from olfactory PNs within the LH. The picture that this paints is of a complex switchboard for olfactory information with many more outputs than we can yet understand. It seems likely that different paths for information flow through the LH may be modulated by external signals such as the internal state of the animal (*Wang et al., 2013*; *Bräcker et al., 2013*; *Sayin et al., 2018*). The next few years should see very rapid progress in understanding the logic of circuits within the LH and their downstream targets through the impact of connectomics approaches combined with the anatomical and functional characterization and tool development that we have begun in this study and *Dolan et al. (2019)*. In conclusion, the *Drosophila* lateral horn now offers a very tractable model to understand the transition between sensory coding and behavior.

## Materials and methods

### Enhancer trap Split-GAL4 screen

We hypothesized the low yield of previous screens to identify LH driver lines was due to a combination of extensive genetic heterogeneity amongst LHNs and the use of classic enhancer trap GAL4 lines, each of which labeled many neuronal classes; if an expression pattern labels many neurons, expression in a small subpopulation may be missed either because they are obscured by brighter

neurons or because neurons of interest do not have a common highly organized structure that observers can more easily discern (*Ito et al., 2003*).

With these concerns in mind, we carried out a Split-GAL4 screen (*Luan et al., 2006*) to generate a more complete and selective set of lines. Split-GAL4 driver lines achieve their increased specificity by the use of two hemidrivers, enhancer trap activation domain lines (ET-AD) and the other for enhancer trap DNA binding domains lines (ET-DBD), each of which must be co-expressed within a cell in order to reconstitute a functional transcription factor. The first stage of our screen was only designed to identify ET-AD and ET-DBD lines that are enriched for LHNs. We dissected 2255 AD hemidrivers and 514 DBD hemidrivers. At this stage we only rejected expression patterns that were either very broad with strong expression across the brain, or contained no labelling at all in the LH. All the lines that passed this basic check (174 DBD hemidriver lines and 282 AD hemidriver lines ) were then stained and imaged at high resolution on a confocal microscope allowing LHNs to be identified and annotated amongst complex expression patterns.

Split-GAL4 screen: DBD and AD enhancers lines were crossed to broadly expressing lines (UAS-CD8-GFP ; UAS-CD8-GFP; elav-AD and UAS-CD8-GFP ; UAS-CD8-GFP; Cha-DBD respectively) and visualized by the expression of mCD8-GFP (*Lee and Luo, 1999*). Lines were selected and annotated based on expression patterns. At the second stage of the screen lines that had similar clusters were crossed and the final expression pattern evaluated and the best lines in terms of specificity and strength of expression were selected for electrophysiology. As we carried out the physiology screen using lines generated in our lab, GMR and VT lines became available for screening. As these lines were sometimes sparse enough to be used directly for physiology we selected some GMR lines for recording as well.

ET-AD insertions were screened by crossing to Cha-DBD (in theory targeting cholinergic excitatory neurons) with a GFP reporter, while ET-DBD insertions were crossed to elav-AD (in theory targeting all neurons). In each case the resulting expression pattern was imaged. Of these lines we chose the best lines based on criteria such as selectivity, and expression strength. The expression pattern was analyzed and annotated for selected lines. Image registration (*Ostrovsky et al., 2013*) to the standard IS2 template brain (*Cachero et al., 2010*; *Manton et al., 2014*) was used to facilitate comparison of lines and clusters. AD and DBD lines that potentially contained the same neurons of interest were then intercrossed to generate more specific lines.

## Hierarchical naming system for LHNs

We chose primary neurite tract as the top level of our hierarchy because each neuron has just one soma and primary neurite tract and because it groups functionally related neurons for example those with common neurotransmitters or similar axonal projections. We named the 31 primary neurite tracts found based on their anterior-posterior and dorso-ventral position with respect to the centre of the LH: AV1-AV7 (AV = anterior ventral), AD1-AD5, PV1-PV12 and PD1-7. Neurons within each tract typically have a shared developmental origin; using co-registered image data (*Yu et al., 2013*; *Ito et al., 2013*; *Manton et al., 2014*), we matched neurons following each of the 31 tracts with 39 parental neuroblasts likely to generate LH neurons (*Figure 1—figure supplement 1A*, this indicates that over a third of the neuronal lineages in the central brain have projections in the LH. Primary neurite tracts were defined using skeletons extracted from light microscopy, and assessing whether co-registered neurons' primary neurites appeared to fasciculate and enter the neuropil together; higher resolution data will likely reveal that some of these tracts can be subdivided.

Primary neurite tracts can be identified in even quite broadly expressed driver lines, but anatomy group distinctions are not always evident and cell types can usually only be convincingly characterized with single neuron images. In our scheme, cell type names are composites incorporating the corresponding tract and anatomy group. As shown in *Figure 2G*, cell type PV5a1 belongs to the posterior ventral tract PV5 and anatomy group PV5a. This provides flexibility for the addition of new cell types, while still ensuring that anatomically and functionally related neurons have similar names; this naming strategy may be useful for other brain areas without clearly defined compartments.

## Computational neuroanatomy

### Neuropil volumes

Using a standard female template brain (*Ito et al., 2014*) we calculated that the first olfactory relay, the AL, has a volume of $1.5 \times 10^5 \ \mu m^3$. Normalizing with respect to the AL, the LH and whole MB occupy relative volumes of 65% and 93%, respectively. However while second order projection neurons leaving the AL make synapses throughout the LH, in the MB they are restricted to the calyx region (relative volume 32% i.e. about half the LH). Similarly, while third order Kenyon cells are completely intrinsic to the MB, LH output neurons have axonal processes outside the LH. Using light level skeleton data (*Chiang et al., 2011*), we find that the on average LHNs have almost exactly the same amount of arbor outside the LH as they have within the LH; note that this calculation was carried out after aggregating by cell type, to avoid cell types that are present at higher frequencies in the FlyCircuit dataset from skewing the results. We therefore conclude that the arbors of third order LHNs are actually likely to occupy a greater volume than MB Kenyon cells (an estimated 130% of the AL volume vs 93%).

On 17th December 2018, the PubMed search ('mushroom body' AND Drosophila) OR ('mushroom bodies' AND Drosophila) returned 1002 results, whereas ('lateral horn' AND Drosophila) OR ('lateral protocerebrum' AND Drosophila) returned 77 results. Note that lateral protocerebrum has sometimes been used as a synonym for lateral horn and on other occasions refers to a wider range of protocerebral neuropils – for this reason it is no longer a recommended term (*Ito et al., 2014*).

### Skeleton data processing pipeline

Open source neuron skeletons were obtained from http://www.flycircuit.tw/ (accessed: January 2017), filtering for any skeleton with processes in within the LH (total: 2245). These skeletons had been automatically reconstructed from sparse image data and the dataset described in previous studies (*Chiang et al., 2011*; *Lee et al., 2012*). A bridging registration (*Manton et al., 2014*) was generated from their Standard Model Brain to our FCWB template brain using the Computational Morphometry Toolkit ( https://www.nitrc.org/projects/cmtk/). Skeletons manually traced from successfully dye-filled neurons (147) during physiological experiments were also registered to a template brain (IS2, *Cachero et al., 2010*) and bridged into the same FCWB space so that all skeletons could be directly compared. Skeletons were then assigned as possible input neurons (1225) to the LH from sensory neuropils or LHNs (1619). A minority (1225) of skeletons seemed to input the LH from other brain areas, for example known MB output neurons (*Aso et al., 2014a*) and others that may be centrifugal inputs from other brain areas. Lacking synaptic data we excluded them from our analysis. Skeletons where split into axonal and dendritic compartments based on a classifier trained on skeleton data from the *Drosophila* medulla (*Lee et al., 2014*) followed by manual editing based on available confocal stack data and expert understanding of neuronal morphology.

Although the axo-dendritic segmentation process was very helpful in defining local vs output LHNs, this was still sometimes challenging. For examples neurons in the AV4 tract, which clearly consists predominantly of LHLNs sometimes project out of the LH to the superior lateral protocerebrum (SLP) (*Figure 2—figure supplement 1B-B'*). Without information about synapse placement it is hard to be certain if these are polarized neurons with axonal arbors in the SLP or local neurons whose domain extends somewhat beyond the anatomically defined LH.

Since the standard LH (*Ito et al., 2014*) is not based solely on PN arborisations and we wanted to exclude neurons that simply passed through the LH making few arborisations outside of their synaptic range. We therefore calculated an 'overlap' score between PN termini within the standard LH neuropil and potential LHN arbor:

$$f(\mathrm{i_s}, j_k) = \sum_{k=1}^{n} e^{-d^2/2\delta^2}$$

Skeletons were resampled so that we considered 'points' in the neuron at 1 μm intervals and an 'overlap score' calculated as the sum of $f(\mathrm{i_s}, j_k)$ over all points $s$ of $i$. Here, $i$ is the axonal portion of a neuron, $j$ is the dendritic portion of a putative target, $\delta$ is the distance between two points at which a synapse might occur (e.g. 1 μm), and $d$ is the euclidean distance between points $s$ and $k$ . The sum was taken of the scores between each point in $i$ and each point in $j$ . Neurons that did not meet a

threshold score of 6000 were excluded as they only skimmed past the PN arbors. Many of the remaining skeletons seemed tangential to the LH but plausibly received direct synaptic input from PNs. A core' set of LHNs was defined using two thresholds, one for overlap score and another for percentage dendrite within the standard LH volume (*Figure 2—figure supplement 1F*).

### Defining supervoxels for LH input and output zones

In order to define overlapping supervoxels that would divide the LH and its output zones into more intuitive anatomical sub-volumes than contiguous isotropic cubes, we first used NBLAST to cluster the axonal and dendritic sub-branches of our LHONs separately. These sub-branches were generated by calculating the Strahler order within the dendrite and removing the highest order segments. We divided these sub-branches each into 25 different clusters. Each of these clusters was then used to generate a supervoxel. For each cluster, a 3-D weighted kernel density estimate was calculated based on points within the clustered sub-branches. Points were placed on the neurites at 1 μm intervals and weighted as 1/total number of points in the cluster, so that supervoxels could be directly compared. An 'inclusion' score for each LHON dendrite, LHLN arbor and PN axon analyzed within each supervoxel was calculated by summing the density estimate for each point in the chosen arbor, again sampled at 1 μm intervals, and normalized by the total number of points in each arbor. A 'projection' score between LH supervoxels and LH target supervoxels was calculated by multiplying the average LH supervoxels and LH target supervoxel inclusion scores for each LHON cell type.

## Immunochemistry and imaging

Immunochemistry was as described previously (*Jefferis et al., 2007* and *Kohl et al., 2013*) except that we used either streptavidin Alexa-568 (ThermoFisher S-11226 1:2000) for the filled neurons with Pacific Blue (ThermoFisher P31582 1:1000) for detection of mouse anti-nc82 or streptavidin Pacific Blue (ThermoFisher S-11222 1:2000) for the filled neurons with Alexa Fluor 568 (ThermoFisher A21144 1:1000) for detection of mouse anti-nc82.

## Electrophysiology and odor stimulation

Electrophysiological recordings were carried out using the general approach of *Wilson et al. (2004)* as modified by *Kohl et al. (2013)*. Briefly, on the day of eclosion flies were $CO_2$ anesthetized and females of the correct genotype were selected. On the day of the experiment (1–2 days later) the fly was cold anesthetized, placed in the recording chamber, and dissected for recording as described in *Kohl et al. (2013)*. Data acquisition was performed as previously described only a Grasshopper 14S5M camera was used and the recording electrodes were 4.5 to 7 MΩ for PNs and 6 to 8 MΩ for LHNs.

Odor stimuli were delivered via a custom odor delivery system (originally described by *Kohl et al., 2013*; see jefferis-lab.org/resources). The setup used for these experiments had a total of 64 channels. Unless otherwise indicated, liquid odors were diluted to 0.2% (2 microliter in 1 ml) of mineral oil (Sigma Aldrich M8410) or distilled water; solid odors were dissolved at 2 mg in 1 ml of solvent. A full list of odors, solvents and dilutions is provided as a supplementary spreadsheet. During stimulus presentation, a portion of the airstream was switched from a solvent control to a selected odorant. The odorized air stream was then mixed with a clean carrier air stream at a 1:8 ratio to give a notional final dilution of $2.5 \times 10^{-4}$. The length of the valve opening stimulus was 250 ms. All the genetic driver line combinations used for electrophysiological recording are given in our supplemental data, driver lines.

## Image analysis

Image registration of nc82 stained brains used CMTK fully automatic intensity-based 3D image registration available at http://www.nitrc.org/projects/cmtk (*Rohlfing and Maurer, 2003*; *Jefferis et al., 2007*). We used the registration parameters and IS2 template brain described in *Cachero et al. (2010)*. Brains from which recordings have been made often have higher background staining in the cortical cell body layer than the IS2 template and sometimes this results in mis-registration. We addressed this issue by using a second template brain consisting of a high background image that had been successfully registered against the IS2 template.

Neuron tracing was carried out in Amira (Thermo Fisher Scientific, Merignac, France) using the hxskeletonize plugin (*Evers et al., 2005*) or with the Simple Neurite Tracer plugin for Fiji/ImageJ (*Longair et al., 2011*). Neurite tracing used Simple Neurite Tracer or the Virtual Finger plugin for Vaa3D (*Peng et al., 2014*) on previously registered image data. Traces were then loaded into R using the nat package (*Jefferis and Manton, 2019*; copy archived at https://github.com/elifescien-ces-publications/nat). When necessary, they were transformed between the space of the JFRC2 and IS2 template brains using the approach of *Manton et al. (2014)* and the nat.flybrains R package (*Manton and Jefferis, 2019*; copy archived at https://github.com/elifesciences-publications/nat.flybrains).

Fine scale analysis of neuronal structure was carried out using NBLAST clustering (*Costa et al., 2016*) as implemented in the nat.nblast R package (*Manton and Jefferis, 2018*; copy archived at https://github.com/elifesciences-publications/nat.nblast); clustering used Ward's method as implemented in the R function hclust.

## Analysis of electrophysiological data

Spike finding was carried out in Igor Pro using the NeuroMatic package (*Rothman and Silver, 2018*) as previously described (*Kohl et al., 2013*). All subsequent analysis was carried out in R using custom, open source packages: gphys (*Jefferis, 2019*; copy archived at https://github.com/elifescien-ces-publications/gphys), physplitdata (*Frechter and Jefferis, 2019b*; copy archived at https://github.com/elifesciences-publications/physplitdata), and physplit.analysis (*Frechter and Jefferis, 2019a*; copy archived at https://github.com/elifesciences-publications/physplit.analysis). Note that to ensure reproducibility, the physplitdata package includes every spike from our study (469 cells, 638602 spikes). We determined if cells showed a significant increase in firing to an odor, by an exact one-sided Poisson test of the number of spikes in windows 0.7–2.2 s after trial onset; we compared odor and control (blank) stimuli using data from four trials per cell (physplit.analysis function poisson-TestOdoursSF). We adjusted raw p values to control the false discovery rate (*Benjamini and Hochberg, 1995*) using R's p.adjust function; responses for a given cell-odor pair were declared significant for FDR adjusted p < 0.01. For single trial response detection we used the same method as above but the responses for a given cell-odor pair were declared significant for FDR adjusted to a slightly more permissive p < 0.04 (single trials necessarily contain less information than the four trials used above). The detection probability for each cell-odor pair was first calculated, then cell response reliability was calculated by averaging across all the significant cell-odor pairs for each cell. Since the weakest significant odor responses (as initially assessed using four trials) necessarily had lower detection probability we also tested the effect of selecting only cell-odor response pairs above a variety of thresholds (*Figure 2—figure supplement 1E*), which resulted in a small increase in response reliability. In the main sequence figure (*Figure 4*), we use a threshold of 5 Hz (which captures 95% of our significant responses).

Odor response profiles for LHNs were initially manually classified, defining a *functional cell type* which was then cross-referenced with other properties. In order for a cell to be included in our population coding analysis it had to have trials for at least 28 odors, spiking responses to at least one odor, in addition to identification of a specific functional cell type.

We also characterized the odor-evoked responses for a given cell-odor pair using peristimulus time histograms (PSTH). The PSTH was calculated for the period 0–3 s using a sliding window of width 500 ms and a time step of 50 ms. We summarized this by the maximum response in the window 0.55–2.4 s (valve opening was from 0.5 s) and when necessary compared this with a baseline spiking rate before odor arrival (calculated for the range 0–0.55 s).

## Odor coding analysis
### Correlation and aggregated correlation heatmaps

To generate the correlation matrix we concatenated the PSTHs for each cell-odor pair to generate one single continuous vector per cell; these vectors were then merged row-wise to form a matrix of all cell odor responses. The cell-odor matrix was then used to calculate the odor response correlation across all cells. For automatic assignment of cell types by physiological or anatomical similarity, we used hierarchical clustering of the NBLAST or odor response similarity matrix using Ward's method as implemented in the R function hclust. We then calculated the percent correct and

Adjusted Rand Index with R package mclust classError and adjustedRandIndex functions for different dendrogram different cut heights; the cut height giving the minimum prediction error was then selected.

Aggregated correlation heatmaps (*Figure 8*) were calculated by generating a mean odor response profile for each cell type and then computing the Pearson cross-correlation coefficient across all these cell types. The correlation shift was calculated by randomizing the odor labels and subtracting the shuffled from the original matrix and taking the mean of the result matrix. This procedure was repeated 1000 times for each group (PNs, LHLNs, LHONs) to generate a distribution.

## ROC analysis

The ROC analysis measured the ability of each functional cell type to categorize the presented odors. We defined the response of each cell type to each odor as the maximum of the baseline subtracted responses in the 6 time bins following odor onset. We then used these responses to compute a separate area under the ROC curve (AUC) score for each cell type as a categorizer for each of the six odor categories. The resulting scores indicated whether the presence of a response by a cell type was an indicator for an odor category. For each cell type we also generated five shuffled responses by randomly permuting the odor labels on the responses. We then computed the maximum AUC scores across odor categories within each cell type, and within each of the shuffles per cell type. Averaging maxima over the shuffles yielded one unshuffled maximum AUC score, and one shuffled one. Finally, we grouped these by the three cell groups (PNs, LHLNs, LHONs), and performed one-sided Mann-Whitney U tests to determine the differences in the median scores.

## Measuring population decoding accuracy

We using linear support vector classifiers to test the population decoding accuracy of each of the three main groups of neurons (PNs, LHLNs, LHONs). The overall procedure is to repeatedly generate random subpopulations of cells of a given size, where each cell is the sole representative of a particular class. We then train linear classifiers to perform identity or category decoding on a trial-by-trial basis for each time bin. The classifiers used have a single parameter that has to be tuned, so we train classifiers for a range of settings of this parameter and store the cross-validated accuracies. We then report the results for the parameter value that maximized the accuracy.

For each run a population of N classes were selected based on a random seed, where N varied from one up to a maximum of 50 (for LHONs). A single cell was then chosen to represent each class. For shuffle trials, the odor labels of the responses were shuffled independently for each cell, bin, and trial. The 120 (30 odors x four trials) responses for each cell in each bin were given numerical labels according to the task (Category or Identity classification). This category label was then assigned to all trials of that class (Category or Identity).

Linear support vector classifiers (SVC) were then trained independently for each bin using the scikit-learn OneVsRestClassifier (https://scikit-learn.org/). Linear SVCs have a single parameter "C' which must be tuned. To perform this tuning we trained classifiers for each of a fixed set of C values, and then determined the best C value (see below). This optimal C value (determined separately for each bin) was then used in the performance plots. The range of C values typically used in practice are powers of 10 spanning a 'sufficiently broad' range. We chose to use a range of $10[-8, -7 \ldots 1]$ as manual inspection showed that the optimal C values were usually within this range, and no consistent plateauing behavior (where the optimal C value appears clipped to either limit of the range of C values used) was observed.

We recorded the cross validated accuracy for each of 4 cross-validation runs as reported by the scikit-learn cross_val_score function. This procedure yields for each bin and each C value, four accuracy values (one from each cross-validation run). We next determine the optimal C value for each bin. To do so, we proceed as follows:

For each time bin of each cell we

1. Computed the mean accuracy after training the support vector classifier over four cross-validations.
2. Split the multiple runs for each time bin x cell into two halves
3. Computed the best C value that is the one with the highest accuracy, using the first half of the data.

4. For each bin of the data in the second half, kept only the result for the best C value for that bin.
5. Computed the mean (traces) and standard deviation (envelopes) of the accuracy over random seeds in the performance plots.

This split procedure reduces the bias associated with selection of the parameter C by measuring performance over independent random samples from those used to tune C. We also compared accuracy of the whole procedure with shuffled data.

## Mapping odor categories to brain regions

We sought to produce a simple visualization that combined the location of LHON axons and dendrites with their odor category selectivity. First each cell type was assigned the odor category for which it was most selective in the ROC analysis. In parallel we calculated the average amount of cable overlap for each cell type the 25 LH input supervoxels and 25 LH output defined earlier. We then combined these two data sources to create a supervoxel-category score by calculating the mean supervoxel score, selecting only the classes whose score was maximal for that category. In this way a high voxel score was generated only when classes were both specific for an odor category and had dense arborization in a specific voxel. We then manually selected two separate thresholds for the LH and the output regions as the two distributions of voxel scores were quite different.

## Electron microscopy data analysis

The whole fly brain EM dataset is described by *Zheng et al. (2018)* and is available for public download at temca2data.org.

## Estimating LHN numbers

We identified the largest primary neurite tracts by combining bridging registrations of existing light level data (*Manton et al., 2014*; *Zheng et al., 2018*) and by simple anatomical tracing. Tract size was calculated by counting all the profiles in a single plane. In this way we identified 17 tracts containing 2465 neuronal profiles. For large tracts, we traced a random subset of these profiles until the first branch point and/or LH entry point – this was used to estimate the number of profiles in the tract belonging to LHNs. The confidence intervals for each tract were calculated assuming that we were sampling from a hypergeometric distribution. Since we only traced 17/31 primary neurite tracts our estimate is a lower bound, but light level data suggest the remaining 14 tracts contain few LHNs (*Figure 2—figure supplement 1F*). We could identify LH output neurons (LHONs) if the first branch point was clearly outside the LH but one of the daughter branches entered the LH. However for those neurons whose first branch point was in the LH it was not possible to determine whether they were LHLNs or LHONs without more extensive tracing. However we were able to confirm the match of EM traced tracts to light level neuronal morphologies by carrying out more extensive tracing of a subset of neurons in each tract. In order to estimate the number of local vs output neurons, we assumed that each tract consisted of its majority cell class (local or output).

## PN to LHN connectivity

Preliminary work in *Zheng et al. (2018)* partially traced and identified most uniglomerular projection neurons. An account of the tracing of projection neuron axons in the LH including marking all their presynapses is given in *Dolan et al. (2018)*. We completed LH arbors for excitatory uniglomerular PNs from the following 51 glomeruli: D, DA1, DA2, DA3, DA4l, DA4m, DC1, DC2, DC3, DC4, DL1, DL2d, DL2v, DL3, DL4, DL5, DM1, DM2, DM3, DM4, DM5, DM6, DP1l, DP1m, V, VA1d, VA1v, VA2, VA3, VA4, VA5, VA6, VA7l, VA7m, VC1, VC2, VC3l, VC3m, VC4, VC5, VL1, VL2a, VL2p, VM1, VM2, VM3, VM4, VM5d, VM5v, VM7d, VM7v.

## Online resources

The source code and data supplements for this study are listed at jefferislab.org/si/frechter18. Dataset and source code packages are hosted on GitHub and archived to zenodo.org. In order to make our cell type annotations for FlyCircuit (*Chiang et al., 2011*) and dye filled skeletons, and our stimulus response data more easily available to the community, we created an interactive R Shiny Web app (shiny.rstudio.com), which can be found at http://jefferislab.org/si/lhlibrary (*Bates and Jefferis,*

*2019c*; copy archived at https://github.com/elifesciences-publications/LHlibrary), based on data collated from several studies, found at https://github.com/jefferislab/lhns (*Bates et al., 2019a*; copy archived at https://github.com/elifesciences-publications/lhns). Skeletons can be viewed within a template brain in 3D, response data can be plotted for specific odors and cell types, and skeletons and response data can be downloaded as SWC and CSV files respectively. Significantly, this 'LH library' also hosts maximal projection images (brain and ventral nervous system), single skeletons from multi-color FlpOut, and 3D vector clouds representing sparse split-GAL4 lines that label LH cell types (from *Dolan et al., 2019*, sister manuscript). The LH library also contains other available datasets that relate to the LH, including PN response data from a calcium imaging study (*Badel et al., 2016*) and functional connectivity data from GH146 uniglomerular PNs providing input to LHNs (*Jeanne et al., 2018*). We were able to cross-match 80/89 3D morphologies reported in *Jeanne et al. (2018)* onto 26 LHN cell types in our data set. Finally we also include predicted connectivity to a wide range of cell types (ASB) to enable the rapid generation of connectivity hypotheses that can be tested functionally or through EM tracing. This web application can also be run directly on an end user's own computer to increase response speed.

## Acknowledgements

We thank G Johnson, P Hasel, B Gyenes, and J Roote for their assistance with Split-GAL4 screening. We thank the LMB workshops for extending our odor delivery device and Jake Grimmett and Toby Darling for assistance with the LMB's compute cluster. We thank R Roberts and P Schlegel for contributing to EM tracing. We gratefully acknowledge the leadership of Ann-Shyn Chiang and his colleagues in sharing the latest version (1.1) of the flycircuit.tw single cell dataset. Images from FlyCircuit were obtained from the NCHC (National Center for High-performance Computing) and NTHU (National Tsing Hua University), Hsinchu, Taiwan. We acknowledge M Landgraf, B White, BJ Dickson, G Rubin, Y Aso and the Bloomington Stock Center and the Developmental Studies Hybridoma Bank for fly stocks and antibodies. We thank R Benton, M Lengyel, G.Turner, J Tuthill, and R Wilson and members of the Jefferis lab for comments on an earlier version of this manuscript. We are grateful to M Lengyel for his input at an early stage of odor coding analysis and J Jeanne and R Wilson for stimulating discussions and sharing results ahead of publication. This work was supported by the Medical Research Council [MRC file reference U105188491], ERC Starting (211089) and Consolidator (649111) grants to GSXEJ, an EMBO Young Investigator and FENS-Kavli Scholar. ASB was supported by the Boehringer Ingelheim Fonds and the Herchel Smith fund at the University of Cambridge. Wellcome Collaborative Award (203261/Z/16/Z) to GSXEJ and DB provided additional support for connectomics.

## Additional information

### Funding

| Funder | Grant reference number | Author |
|---|---|---|
| European Commission | ERC CoG 649111 | Shahar Frechter<br>Alexander Shakeel Bates<br>Sina Tootoonian<br>Michael-John Dolan<br>Gregory Jefferis |
| Medical Research Council | U105188491 | Shahar Frechter<br>Alexander Shakeel Bates<br>Michael-John Dolan<br>James Manton<br>Johannes Kohl<br>Gregory Jefferis |
| European Commission | ERC StG 211089 | Shahar Frechter<br>Michael-John Dolan<br>James Manton<br>Gregory Jefferis |

| Wellcome | 203261/Z/16/Z | Arian Rokkum Jamasb<br>Davi Bock<br>Gregory Jefferis |

The funders had no role in study design, data collection and interpretation, or the decision to submit the work for publication.

## Author contributions

Shahar Frechter, Conceptualization, Resources, Data curation, Formal analysis, Supervision, Investigation, Methodology, Writing—original draft, Project administration, Writing—review and editing; Alexander Shakeel Bates, Writing—original draft, Writing—review and editing, Formal analysis, Investigation, Methodology, Data curation; Sina Tootoonian, Formal analysis, Writing—original draft; Michael-John Dolan, Resources, Data curation; James Manton, Data curation, Software; Arian Rokkum Jamasb, Johannes Kohl, Resources, Investigation; Davi Bock, Resources, Supervision, Funding acquisition; Gregory Jefferis, Conceptualization, Software, Formal analysis, Supervision, Funding acquisition, Investigation, Visualization, Methodology, Writing—original draft, Writing—review and editing

## Author ORCIDs

Shahar Frechter https://orcid.org/0000-0002-0431-5849
Alexander Shakeel Bates https://orcid.org/0000-0002-1195-0445
Sina Tootoonian https://orcid.org/0000-0002-3990-8724
Michael-John Dolan https://orcid.org/0000-0001-9666-3682
James Manton http://orcid.org/0000-0001-9260-3156
Arian Rokkum Jamasb http://orcid.org/0000-0002-6727-7579
Davi Bock http://orcid.org/0000-0002-8218-7926
Gregory Jefferis https://orcid.org/0000-0002-0587-9355

## Decision letter and Author response

Decision letter https://doi.org/10.7554/eLife.44590.025
Author response https://doi.org/10.7554/eLife.44590.026

# Additional files

## Supplementary files

• Supplementary file 1. Summary odor response data for recorded neurons expressed as baseline subtracted spike rate (Hz).
DOI: https://doi.org/10.7554/eLife.44590.018

• Supplementary file 2. The drive lines used in this study to target specific populations of neurons. More specific driver have now been reported our sister paper, *Dolan et al. (2019)*.
DOI: https://doi.org/10.7554/eLife.44590.019

• Supplementary file 3. A zip file containing SWC format neuronal skeletons.
DOI: https://doi.org/10.7554/eLife.44590.020

• Supplementary file 4. The metadata associated with each skeleton, including cell type annotations.
DOI: https://doi.org/10.7554/eLife.44590.021

• Supplementary file 5. The odors set used in our study, including details of their chemical categories and their CAS numbers.
DOI: https://doi.org/10.7554/eLife.44590.022

• Supplementary file 6. Summary of the EM connectivity between the EM reconstructed PNs and LHNs in *Figure 10*.
DOI: https://doi.org/10.7554/eLife.44590.023

## Data availability

Digital skeletons for neuronal morphology and summary electrophysiological data have been provided as supplemental zip files. An interactive version of these data is available online and full source

data and source code are available on GitHub and archived at zenodo. Full links for these materials are provided in Materials and Methods and can also be found by following links at http://jefferislab.org/si/frechter18.

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
