## [Decision Letter]

Thank you for submitting your article "Functional and anatomical specificity in a higher olfactory centre" for consideration by *eLife*. Your article has been reviewed by two peer reviewers, and the evaluation has been overseen by K VijayRaghavan as the Senior and Reviewing Editor. The following individual involved in review of your submission has agreed to reveal her identity: Ilona C Grunwald Kadow (Reviewer #2).

The reviewers have discussed the reviews with one another and the Reviewing Editor has drafted this decision to help you prepare a revised submission.

Summary:

Frechter et al. present a comprehensive and impressive dataset characterizing a brain center that for multiple reasons has been 'dismissed' for further analysis from many studies dealing with chemosensory processing in insects and in particular in *Drosophila*. As the authors point out (i.e., 'This large investment in neuropil volume argues for the significance of the LH in sensory processing, whereas the literature currently contains 13 fold more studies of the *Drosophila* MB than LH'), the LH is far understudied and this present work together with its companion manuscript (Dolan et al., 2019 *eLife*) will contribute significantly to changing this imbalance of knowledge.

The authors conclude that the LH shows highly stereotyped odor responses, which can be best described as responses to odor categories rather than odor valence. Furthermore, comparing the LH to the cortical amygdala reveals that these two brain regions encode innate information in a different form.

(Please see the detailed comments below. Additional analysis as suggested would be very interesting but not essential at this stage given that the manuscript is already quite voluminous.

The authors should address key points in a rebuttal, carry out additional analysis only if compatible with a speedy publication, and invest a time into their figures and figure legends.)

The authors' main findings/achievements are. These are listed so that the comments that follow can be seen in this perspective.

A) Anatomy/tools:

1) Using EM tracing techniques, they estimate that the number of LHNs (~1400) is significantly larger than expected from previous estimates.

2) Genetic screening of a couple of thousand lines allows them to identify 234 split-Gal4 lines for labeling specific subsets of LHNs. They further identify 69 different subgroups of LHNs based on these and other lines and a three-level system of anatomical categories.

3) Based on these lines, they generated a collection of specific Split-Gal4 lines for further characterization (see also sister manuscript Dolan et al.)

4) In a next step, they combine different neuroanatomy databases to further subcategorize cell types based on morphology. For instance, they define intrinsic LHNs, LH output neurons, different projection zones including shared zones with MB output etc. They also make all this data is available and searchable online (the link did not work for me, however…)!

B) Type:

1) They record electro-physiologically from ~600 cells including LHINs, LHONs, PNs and find that LHNs have higher input resistance and lower capacitance as compared to PNs; a finding not only interesting in regard to their function/coding, but also of relevance for other questions such as lifetime, metabolism of PNs vs. LHNs.

2) Another important finding is that LHONs respond to ~3x more odors than PNs.

3) The authors used different criteria to try to define cell types based on anatomy and/or function. They used automated clustering analysis and found that many cell types fell into the same odor response type and anatomical (at least on the level of the primary neurite morphology) categories that they had defined by hand.

4) Based on odor responses they define 33 cell types with stereotyped responses, and argue that this type of response stereotopy is likely also true for the remaining anatomical classes with insufficient response information.

C) Coding/function:

1) Prior work hypothesized that the LH might code for valence. Given the observed response profiles as well as the important aspect of behavioral context, the authors argue that 'good vs. bad' are unlikely categories for LH odor coding. Instead, they test several other hypotheses including chemical features.

2) They find that ~70% of LHONs respond by odor categories such as amine vs. acetate or pheromones.

3) This type of categorization held true also for LHON targets in other brain regions suggesting segregating pathways depending on odor category/behavioral context.

4) Using the whole brain EM data set by Davi Bock and colleagues, they map out connections between uniglomerular PNs and LHNs and find that ~5 PNs input to every LHN, which is interestingly similar for KC input!

5) However, in contrast to KC input, LHON/PN synapses were much more numerous and LHONs, in addition, received significantly more input from other local neurons.

Major Comments:

1) Overall, this is a well written, impressive manuscript. It is obvious that much more analysis could be done in the future, and that this work will open the way for many different studies including behavior and circuit mapping. Together with its sister manuscript (Dolan et al., 2019 *eLife*), this work builds an extremely powerful, comprehensive, and informative package that should appear ideally back-to-back. In fact, as Aso et al., 2014 a and b, have done for the MB, these two manuscripts will provide a strong push for the LH.

2) There are concerns regarding figures and figure legends. In the figures, most writing is too small. While we understand that this is difficult to change, we would encourage the authors to think of ways to put additional labeling (for instance of some of the axes) to just indicate what kind of more detailed information is 'hidden' in small print. Similarly, short figure legends can be great, but once again we would have greatly appreciated some more details in the legends, which could even include some of the 'take-home' messages of the particular panel or figure. Given the breadth and length of the manuscript, a reader is otherwise forced to jump back-and-forth between main text, legend, and Materials and methods constantly. And most readers are not ready to spend a lot of time to read a paper. In addition, many of the links provided by the authors did not work, please check this.

3) The authors discuss the hypothesis that the mushroom body is for odor identification while the LH is for odor evaluation/classification. This could potentially be very nicely tested with their existing data, by asking: how good is the population of LHN recordings at identifying odors? – i.e., is it worse than PNs? Possible metrics:

a) Figure 7 looks at correlations in odor response profiles between different LHON, LHLN and PN types. Could they reverse this and instead look at correlation in population responses, between different odors? Are population responses more correlated for LHNs than for PNs? (Of course, this would need similar controls as the rest of Figure 7 to ensure that any difference isn't an artifact of the number of cell types recorded).

b) Cluster analysis based on individual trials – how well separated are the clusters corresponding to each individual odor? How many errors are made when identifying which odor is a single trial? (as in Hige et al. Nature 2015 – where MBONs performed worse than KCs) (confounding factor here is that LHN responses are more reliable than PNs – but if LHNs are *worse* than PNs at clustering, then the confound is in the other direction)

c) ROC/AUC analysis like they do in Figure 8 for odor categories.

4) Can the authors calculate lifetime and population sparseness for LHLNs, LHONs, and PNs (as an additional measure beyond just% of odors with significant responses)?

5) We have concerns about the calculation of LH neuropil volume. They calculate the total neuropil volume of the LH, then multiply by a factor of 2 on the basis that on average, LHNs have the same amount of arbor outside the LH as inside. This is based on light level skeleton data which I'm guessing reveals length but not volume. What if LHNs have thinner axons outside the LH than inside? Also, a distribution concern: What if LHNs with very short axons have a greater proportion of the axon outside the LH, while LHNs with very long axons have a smaller proportion outside the LH? Then on average LHNs would have the same amount of arbor inside as outside the LH, yet the total amount of neuropil outside the LH would be smaller than the total amount inside. Can the authors rule out these possibilities or place some confidence bounds on their estimate?

6) The authors use the Rand index of ~ 0.6 to say that their manual classification is "well-grounded" – but we don't have any intuition for whether a Rand index of 0.6 is "good". Would they also have said their classification was "well-grounded" if the Rand index was, say, 0.2? (i.e. is this basically an unfalsifiable statement) Can this be benchmarked against some examples where the reader can have some intuitive grasp?

---

## [Author Response]

Major Comments:1) Overall, this is a well written, impressive manuscript. It is obvious that much more analysis could be done in the future, and that this work will open the way for many different studies including behavior and circuit mapping. Together with its sister manuscript (Dolan et al., 2019 eLife), this work builds an extremely powerful, comprehensive, and informative package that should appear ideally back-to-back. In fact, as Aso et al., 2014 a and b, have done for the MB, these two manuscripts will provide a strong push for the LH.

We thank the reviewers for their careful and positive evaluation of this paper.

2) There are concerns regarding figures and figure legends. In the figures, most writing is too small. While we understand that this is difficult to change, we would encourage the authors to think of ways to put additional labeling (for instance of some of the axes) to just indicate what kind of more detailed information is 'hidden' in small print. Similarly, short figure legends can be great, but once again we would have greatly appreciated some more details in the legends, which could even include some of the 'take-home' messages of the particular panel or figure. Given the breadth and length of the manuscript, a reader is otherwise forced to jump back-and-forth between main text, legend, and Materials and methods constantly. And most readers are not ready to spend a lot of time to read a paper. In addition, many of the links provided by the authors did not work, please check this.

We thank the reviewers for their suggestions and hope that our manuscript is now more readable as a result of changes that we have made. These changes include:

We have split our original Figure 3 in two to allow the size of the summary figures to be increased.

We have likewise split our original Figure 8 in order to ensure legibility while adding additional panels to address reviewer point 3.

We have added some additional axis labeling (e.g. Figure 1E to show what kind of information is in the X axis) and figure titles (e.g. Figure 6).

We have added additional explanatory text to figure legends (Figure 4, 5) We have added a missing scale bar to Figure 5.

3) The authors discuss the hypothesis that the mushroom body is for odor identification while the LH is for odor evaluation/classification. This could potentially be very nicely tested with their existing data, by asking: how good is the population of LHN recordings at identifying odors? – i.e., is it worse than PNs? Possible metrics:a) Figure 7 looks at correlations in odor response profiles between different LHON, LHLN and PN types. Could they reverse this and instead look at correlation in population responses, between different odors? Are population responses more correlated for LHNs than for PNs? (Of course, this would need similar controls as the rest of Figure 7 to ensure that any difference isn't an artifact of the number of cell types recorded).b) Cluster analysis based on individual trials – how well separated are the clusters corresponding to each individual odor? How many errors are made when identifying which odor is a single trial? (as in Hige et al. Nature 2015 – where MBONs performed worse than KCs) (confounding factor here is that LHN responses are more reliable than PNs – but if LHNs are worse than PNs at clustering, then the confound is in the other direction)c) ROC/AUC analysis like they do in Figure 8 for odor categories.

We thank the reviewers for this question, which has pushed us to add an interesting additional analysis to the paper. We considered all three of the metrics that they suggested (including carrying out the analysis in a and corresponding with Glenn Turner to verify the details of the analysis in b). However in the end, we concluded that none of these suggestions were effective ways to address the underlying question, “How good is the LHN vs. PN population at identifying odors?” for our data.

Instead we adopted an approach similar to the one used by Bhandawat et al., 2007. We used a linear classifier (specifically a support vector classifier) to predict odour identity from neuronal population responses. We constructed a virtual population of PNs or LHONs by selecting a single cell for each cell type. We then trained the support vector classifier to predict odor identity and tested its performance using a leave one out cross-validation step. Bootstrapping the selection of different neurons gave an indication of variability in this performance. We also tested the impact of varying the number of cell types on the prediction accuracy. These results are included at the end of Figure 9 (formerly Figure 8) and described in the text (subsection “Encoding of odor categories”, fifth paragraph). Additional methodological details are provided (Materials and methods subsection “Measuring population decoding accuracy”).

The summary of these new results is that LHONs showed a consistent advantage in odour categorisation for cell populations as well as single cells. However for odor identification, LHON populations (but not single LHONs) outperform PNs, which is probably not the reviewer’s prediction. The likely reason for this is again the tuning breadth of LHONs – they respond to more odours (by integrating multiple input channels). Effectively for a given number of cell types an LHN population has more information than a PN population – indeed some of our test odours almost never produced a significant PN response, while still exciting numerous LHNs. All of this forces us to the conclusion that we mention in the text – that a better understanding of this relationship may need to await an improved understanding of the odour channels that LHONs actually integrate. Forthcoming connectomics data means that within the next 1-2 years we should have such data.

Now returning to the initial motivation. As the reviewers noted, we discussed the “hypothesis that the mushroom body is for odor identification while the LH is for odor evaluation/classification”. We do indeed favour a version of this hypothesis, concluding our discussion of this point (emphasis added):

“Nevertheless synthesizing the results in this study with other new work (Dolan et al., 2018a,b; Huoviala et al., 2018; Jeanne et al., 2018) does support the hypothesis that stereotyped integration in the LH could support genetically determined categorical odor representations, while the MB may enable *identification of specific learned* odors.”

However we would also note that we do not feel that it makes a strong prediction about the relative performance of PNs vs LHONs in odor identification. It does make a stronger prediction about relative odor identification performance of LHONs vs Kenyon cells, but that is not something that we can test at this point – it would odour response data for a large KC population with our stimulus conditions.4) Can the authors calculate lifetime and population sparseness for LHLNs, LHONs, and PNs? (as an additional measure beyond just% of odors with significant responses).

We have added lifetime and population sparseness measures to the new Figure 4 (formerly the lower half of Figure 3) and reference these panels in the text (Results subsection “Odor responses of lateral horn neurons”, fourth paragraph). These results follow the anticipated trends i.e. decreasing sparseness for PNs, LHLNs, LHONs.

5) We have concerns about the calculation of LH neuropil volume. They calculate the total neuropil volume of the LH, then multiply by a factor of 2 on the basis that on average, LHNs have the same amount of arbor outside the LH as inside. This is based on light level skeleton data which I'm guessing reveals length but not volume. What if LHNs have thinner axons outside the LH than inside?

This an interesting point. We have recently measured the volume to cable length ratio of axonal and dendritic arbours for a sample of LHONs (using as yet unpublished auto-segmentation data generated by Peter Li and collaborators, available in preprint form at https://doi.org/10.1101/605634); we find that the axonal arbours have ~70% larger volume:length ratio than dendrites on average. Therefore it seems that LHN axons are thicker than their dendrites and we can reassure the reviewers that this concern does not hold.

Also, a distribution concern: What if LHNs with very short axons have a greater proportion of the axon outside the LH, while LHNs with very long axons have a smaller proportion outside the LH? Then on average LHNs would have the same amount of arbor inside as outside the LH, yet the total amount of neuropil outside the LH would be smaller than the total amount inside. Can the authors rule out these possibilities or place some confidence bounds on their estimate?

We summed the total arbour volume outside the LH for one sample flycircuit neuron for each of 167 cell types and compared it with the summed arbour volume inside the LH for the selected neurons. The particular distributional concern mentioned by the reviewers therefore does not apply, since we divided the two summed volumes rather than calculating the mean of 167 cable ratios. Of course we cannot exclude the possibility that neurons with extensive arbours outside the LH may be overrepresented in flycircuit, but we have taken a large sample and we only report our figures one figures to 1 s.f. and make it clear that this is an estimate.

We therefore think that our statement “we estimate that the total volume of LHN arbors is therefore actually 40% greater than the MB” remains a reasonable summary.

6) The authors use the Rand index of ~ 0.6 to say that their manual classification is "well-grounded" – but we don't have any intuition for whether a Rand index of 0.6 is "good". Would they also have said their classification was "well-grounded" if the Rand index was, say, 0.2? (i.e. is this basically an unfalsifiable statement) Can this be benchmarked against some examples where the reader can have some intuitive grasp?

We agree that few readers may have any intuition about what constitutes a “good” value of the adjusted Rand index. The adjusted Rand index does show that the classification accuracy is well above chance (since the adjusted Rand index would be 0 at chance performance); we have pointed this out in the text (subsection “Fine scale anatomical clustering confirms LHN classification”, second paragraph). However we now also report in the text (third paragraph of the aforementioned subsection) the final proportion correctly classified by anatomy and relate this to Figure 7E, which shows excellent agreement between the NBLAST anatomical clustering and functional cell types.